# CERBERUS: MINIMALISTIC MULTI-SHARD BYZANTINE-RESILIENT TRANSACTION PROCESSING

JELLE HELLINGS
*Department of Computing and Software*
*McMaster University*
*jhellings@mcmaster.ca*

DANIEL P. HUGHES
*RDX Works Ltd*
*dan@rdx.works*

JOSHUA PRIMERO
*RDX Works Ltd*
*josh@rdx.works*

MOHAMMAD SADOGHI
*Exploratory Systems Lab*
*Department of Computer Science*
*University of California, Davis*
*msadoghi@ucdavis.edu*

## Abstract

To enable scalable resilient blockchain systems, several powerful general-purpose approaches toward sharding such systems have been demonstrated. Unfortunately, these approaches all come with substantial costs for ordering and execution of multi-shard transactions.

In this work, we ask whether one can achieve significant cost reductions for processing multi-shard transactions by limiting the type of workloads supported. To initiate the study of this problem, we propose CERBERUS, a family of minimalistic primitives for processing single-shard and multi-shard UTXO-like transactions. The first CERBERUS variant we propose is core-CERBERUS (CCERBERUS). CCERBERUS uses *strict UTXO-based environmental requirements* to enable powerful multi-shard transaction processing with an absolute minimum amount of coordination between shards. In the environment we designed CCERBERUS for, CCERBERUS will operate *perfectly* with respect to all transactions proposed and approved by well-behaved clients, but does not provide any other guarantees.

To illustrate that CCERBERUS-like protocols can also be of use in environments with faulty clients, we also demonstrate *two* generalizations of CCERBERUS, *optimistic*-CERBERUS and *resilient*-CERBERUS, that make different tradeoffs in complexity and costs when dealing with faulty behavior and attacks. Finally, we compare these three protocols and show their potential scalability and performance benefits over state-of-the-art general-purpose systems. These results underline the importance of the study of specialized approaches toward sharding in resilient systems.

## 1 Introduction

The advent of blockchain applications and technology has rejuvenated interest of companies, governments, and developers in resilient distributed fully-replicated systems and the distributed ledger technology (DLT) that powers them. Indeed, in the last decade we have seen a surge of interest in reimagining systems and build them using DLT networks. Examples can be found in the financial and banking sector [15, 42, 57], IoT [48], health care [29, 43], supply chain tracking, advertising, and in databases [5, 23, 31, 32, 53–55]. This wide interest is easily explained, as blockchains promise to improve resilience against both failures and malicious behavior, while enabling the federated management of data by many participants.

To illustrate this, we look at the financial sector. Current traditional banking infrastructure is often rigid, slow, and creates substantial frictional costs. It is estimated that the yearly cost of transactional friction alone is \$71 billion [8] in the financial sector, creating a strong desire for alternatives. This sector is a perfect match for DLT, as it enables systems that manage digital assets and financial transactions in more flexible, fast, and open federated infrastructures that eliminate the friction caused by individual private databases maintained by banks and financial services providers. Consequently, it is expected that a large part of the financial sector will move towards DLT [18].

At the core of DLT is the *replicated state* maintained by the network in the form of a ledger of transactions. In traditional blockchains, this ledger is fully replicated among all participants using consensus protocols [14,31,40,48,52]. For many practical use-cases, one can choose to use either permissionless consensus solutions that are operated via economic self-incentivization through cryptocurrencies (e.g., Nakamoto consensus [51, 62]), or permissioned consensus solutions that require vetted participation (e.g, PBFT, POE, and HOTSTUFF [16, 33, 64]). Unfortunately, the design of consensus protocols are severely limited in their ability to provide the *high transaction throughput* that is needed to address practical needs, e.g., in the financial sector. Indeed, on the one hand, we see that permissionless solutions can easily scale to thousands of participants, but are severely limited in their transaction processing throughput. For example, in Ethereum, a popular public permissionless DLT platform, the rapid growth of decentralized finance applications [12] causes its network fees to rise precipitously as participants bid for limited network capacity [7], while Bitcoin can only process a few transactions per second [57]. On the other hand, solutions that use permissioned consensus protocols such as PBFT can reach much higher throughput. Typically, these solutions are still fully-replicated resilient systems, however. Hence, the speed by which individual replicas can process transactions provides an upper-bound on the performance of these solutions, ruling out scalability. Furthermore, adding replicas will actively decrease performance of these solutions, as full replication among more replicas increases the cost of full replication (e.g., via consensus). As such, these solutions lack the scalability required by many modern high-throughput data-based

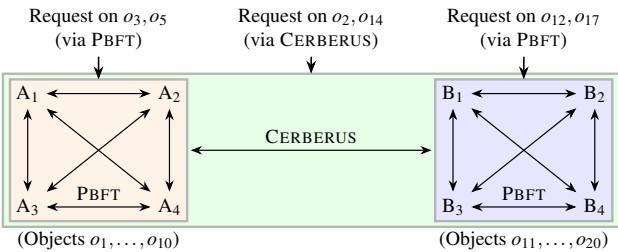

Figure 1: A *sharded* design in which two resilient blockchains each hold only a part of the data. If a transaction only affects objects within a single resilient blockchain (cluster), then the transaction can be processed via a *local decision* within that cluster via *traditional* PBFT *consensus*, whereas multi-shard transactions are processed via CERBERUS (proposed in this work).

applications [31, 34, 55].

Recently, several general-purpose consensus-based systems have been proposed that use *sharding* to combat the limitations of fully-replicated consensus-based systems [1, 3, 4, 17, 36, 38, 58]. In these systems, one partitions the data among several *shards* that each can potentially operate mostly-independent on their data, while only requiring inter-shard coordination to process multi-shard transactions that affect data on several shards (see Figure 1).

The choice of protocol for such multi-shard transaction processing determines greatly the scalability benefits of sharding and the overhead costs incurred by sharding, however [1, 3, 4, 17, 36, 38, 58]. In practice, existing proposals for sharding consensus-based systems have taken a general-purpose approach aiming at serving any workload. Unfortunately, such genericity comes at a cost, and existing proposals either have high coordination costs, incur high latencies, or have severe bottlenecks with multi-shard workloads.

In this work, we ask whether one can improve on the state-of-the-art proposals for sharded consensus-based resilient systems by *limiting* the type of workloads supported by the systems. In specific, we propose the following problem for further study:

> **Problem.** Can one *reduce* the cost of coordination in the design of sharded consensus-based systems by limiting the types of workloads supported?

In this paper, we give a preliminary *positive* answer for the above problem. In specific, in this paper we limit the types of workloads to UTXO-transactions and we use properties of these UTXO-transactions to design the CERBERUS family of *minimalistic* multi-shard transaction processing protocols: by using the properties of UTXO-transactions to our advantage, the CERBERUS family of multi-shard transaction processing protocols are able to reduce coordination, e.g., in terms of local consensus steps or in terms of communication between

shards, to a minimum. To be able to adapt to the needs of specific use-cases, we propose three variants of CERBERUS:

1. In Section 4, we propose Core-CERBERUS (CCERBERUS), a design specialized for processing *UTXO-like transactions*. CCERBERUS uses strict environmental assumptions on UTXO-transactions to its advantage to yield a *minimalistic* design that only requires a *single local consensus step* in affected shards, an absolute minimum. Furthermore, CCERBERUS requires only a single round of information sharing between shards. This information sharing can be implemented either via an all-to-all communication step (favoring latency over bandwidth usage) or via an all-to-one-to-all communication step (favoring bandwidth usage over latency).

   Even with this minimalistic design, CCERBERUS will operate *perfectly* with respect to all transactions proposed and approved by well-behaved clients: CCERBERUS is able to correctly execute all such transactions, even in the presence of transactions proposed by malicious clients (CCERBERUS may fail to process transactions originating from malicious clients, however).

To also support more general-purpose environments in which clients are malicious or can legitimately approve conflicting transactions, we propose Optimistic-CERBERUS and Resilient-CERBERUS, *two* generalizations of CCERBERUS that each deal with the strict environmental assumptions of CCERBERUS while preserving the minimalistic design of CCERBERUS:

2. In Section 5, we propose Optimistic-CERBERUS. In the design of Optimistic-CERBERUS (OCERBERUS), we assume that malicious behavior is rare and we optimize the normal-case operations. We do so by keeping the normal-case operations as minimalistic as possible by utilizing a *single* multi-shard consensus step to execute multi-shard transactions in the normal case.

   This multi-shard consensus step combines the local consensus steps of CCERBERUS and the information sharing steps of CCERBERUS into a single step. As with CCERBERUS, this step can either favor latency or bandwidth. When compared to CCERBERUS, the multi-shard consensus step does not require any additional coordination phases in the well-behaved optimistic case, while still being able to lift the environmental assumptions of CCERBERUS and lowering the latency of transaction processing in most cases. In doing so, OCERBERUS does require intricate coordination when recovering from attacks, however.

3. In Section 6, we propose Resilient-CERBERUS. In the design of Resilient-CERBERUS, we assume that malicious behavior is common and we add sufficient coordination

to the normal-case operations of CCERBERUS to enable a simpler recovery path that can deal with malicious replicas within a shard without interfering with the operations of any other shards, allowing PCERBERUS to operate in a general-purpose fault-tolerant environments without significant costs to recover from attacks.

In Section 7, we show that all three variants of CERBERUS provide strong ordering guarantees based on their usage of UTXO-transactions. Finally, in Section 8, we compare the three CERBERUS protocols and show their potential scalability and performance benefits over state-of-the-art general-purpose systems

## 2  Preliminaries

As permissioned blockchains already have much higher throughputs than permissionless blockchains, we will focus on permissioned blockchains in this paper.

First, we introduce the system model, the sharding model, the data model, the transaction model, and the terminology and notation used throughout this paper.

If $S$ is a set of replicas, then $\mathcal{G}(S)$ denotes the non-faulty *good replicas* in $S$ that always operate as intended, and $\mathcal{F}(S) = S \setminus \mathcal{G}(S)$ denotes the remaining replicas in $S$ that are *faulty* and can act *Byzantine*, deviate from the intended operations, or even operate in coordinated malicious manners. We write $\mathbf{n}_S = |S|$, $\mathbf{g}_S = |\mathcal{G}(S)|$, and $\mathbf{f}_S = |S \setminus \mathcal{G}(S)| = \mathbf{n}_S - \mathbf{g}_S$ to denote the number of replicas in $S$, good replicas in $S$, and faulty replicas in $S$, respectively.

We assume that communication between replicas is *authenticated*: on receipt of a message $m$ from replica $\mathsf{R} \in \mathfrak{R}$, one can determine that $\mathsf{R}$ did sent $m$ if $\mathsf{R} \in \mathcal{G}(\mathfrak{R})$. Hence, faulty replicas are able to impersonate each other, but are not able to impersonate good replicas. To provide authenticated communication under practical assumptions, we can rely on cryptographic primitives such as digital signatures and threshold signatures [44, 59].

Let $\mathfrak{R}$ be a set of replicas. In a *sharded fault-tolerant system* over $\mathfrak{R}$, the replicas are partitioned into sets $\mathrm{shards}(\mathfrak{R}) = \{\mathcal{S}_0, \ldots, \mathcal{S}_\mathbf{z}\}$ such that the replicas in $\mathcal{S}_i$, $0 \leq i \leq \mathbf{z}$, operate as an independent Byzantine fault-tolerant system. As each $\mathcal{S}_i$ operates as an independent Byzantine fault-tolerant system, we require $\mathbf{n}_{\mathcal{S}_i} > 3\mathbf{f}_{\mathcal{S}_i}$, a minimal requirement to enable Byzantine fault-tolerance in an asynchronous environment [20, 21]. We assume that every shard $\mathcal{S} \in \mathrm{shards}(\mathfrak{R})$ has a unique identifier $\mathrm{id}(\mathcal{S})$.

**Assumption 2.1.** We assume *adversaries* that can, at will, choose and control any replica $\mathsf{R} \in \mathcal{S}$ in any shard $\mathcal{S} \in \mathrm{shards}(\mathfrak{R})$ in the sharded fault-tolerant system as long as, for each shard $\mathcal{S}' \in \mathrm{shards}(\mathfrak{R})$, the adversaries only control up to $\mathbf{f}_{\mathcal{S}'}$ replicas in $\mathcal{S}'$.

We use the *object-dataset model* in which data is modeled as a collection of *objects*. Each object $o$ has a unique *identifier* $\mathrm{id}(o)$ and a unique *owner* $\mathrm{owner}(o)$. In this setting, the owner of an object typically is some client that can authenticate transactions involving that object. For example, if objects represent monetary tokens, then the owner can approve transfers of these tokens.

In the following, we assume that all owners are *clients* of the system that manages these objects. The only operations that one can perform on an object are *construction* and *destruction*. An object cannot be recreated, as the attempted recreation of an object $o$ will result in a new object $o'$ with a distinct identifier $(\mathrm{id}(o) \neq \mathrm{id}(o'))$.

Changes to object-dataset data are made via transactions requested by clients. We write $\langle \tau \rangle_c$ to denote a transaction $\tau$ requested by a client $c$. We assume that all transactions are *UTXO-like transactions*: a transaction $\tau$ first produces resources by destructing a set of *input objects* and then consumes these resources in the construction of a set of *output objects*. We do not rely on the exact rules regarding the production and consumption of resources, as they are highly application-specific. Given a transaction $\tau$, we write $\mathrm{Inputs}(\tau)$ and $\mathrm{Outputs}(\tau)$ to denote the input objects and output objects of $\tau$, respectively, and we write $\mathrm{Objects}(\tau) = \mathrm{Inputs}(\tau) \cup \mathrm{Outputs}(\tau)$.

UTXO-like transactions are mainly known for their usage in Bitcoin and other cryptocurrencies [51]. They form an abstract model that can easily be used for many other types of workloads. Next, we consider three types of applications in which we can use UTXO-like transactions.

*Example* 2.2. The traditional example would be *banking*: objects representing unique tokens (e.g., coins or other unique objects) that can be exchanged between owners. For example, if Ana owns tokens $\{232, 437, 1211, 1234\}$ and Bo owns tokens $\{11, 124, 423\}$, they can work together to send two tokens to Eva:

$$\tau := \mathrm{Inputs}: \{1234, 423\} \to$$
$$\mathrm{Outputs}: \{1244 \mapsto \mathrm{Eva}, 9823 \mapsto \mathrm{Eva}\}.$$

Transaction $\tau$ needs to be signed by Ana and Bo to express their approval. In this example, 1244 and 9823 are new unique identifiers (the way in which these identifiers are generated will depend on the specific requirements of the application. e.g., they can be based on a digest of the transaction).

To further underline the flexibility of UTXO-transactions, we will next consider two applications outside the traditional banking setting.

*Example* 2.3. Consider a game inventory in which objects (e.g., iron and wood) are owned by users. In this case, a transaction can be (1) the trade of objects between users; and (2) the construction of new objects out of their building blocks. For example, a setting in which Ana owns objects $\{\mathrm{Iron}_{123}, \mathrm{Iron}_{125}\}$ and Bo owns $\{\mathrm{Wood}_{15}\}$. A trade transac-

tion would look like:

$$\tau_1 := \text{Inputs:}\{\text{Iron}_{123}, \text{Wood}_{15}\} \rightarrow$$
$$\text{Outputs:}\{\text{Iron}_{879} \mapsto \text{Bo}, \text{Wood}_{321} \mapsto \text{Ana}\}.$$

Again, this transaction needs to be signed by Ana and Bo to express their approval. After execution of this transaction, Ana will own $\{\text{Iron}_{125}, \text{Wood}_{321}\}$ after which she can construct an axe from the building blocks she owns and gift this axe to Eva:

$$\tau_2 := \text{Inputs:}\{\text{Iron}_{125}, \text{Wood}_{321}\} \rightarrow$$
$$\text{Outputs:}\{\text{Axe}_{91} \mapsto \text{Eva}\}.$$

This transaction needs to be signed by Ana for her approval. After execution of this transaction, Ana will not own anything, while Eva owns $\{\text{Axe}_{91}\}$.

*Example* 2.4. As a last example, we consider how a versioned file system can be represented via UTXO-like transactions: each object is a file (owned by some user) and transactions represent one or more changes to a file. In this model, normal file edits turn a file object into a new object representing the new version of that file (typically owned by the same user, but ownership can be transferred). In such a versioned file system, new files can be created (transactions with only outputs), existing files can be updated (transactions with equal amounts of inputs and outputs) or removed (transactions with only inputs), files can be duplicated (transactions with more outputs than inputs), and files can be merged (transactions with more inputs than outputs). Notice that in a versioned file system, updating or removing a file only impacts a *new version* of the file system in which the affected file is updated or removed: these file changes do not affect the previous versions of the file (which typically remain available via a *ledger* that keeps track of all transactions executed by the system).

**Assumption 2.5.** Given a transaction $\tau$, we assume that one can determine $\text{Inputs}(\tau)$ and $\text{Outputs}(\tau)$ a-priori.

To simplify presentation, we assume throughout this paper that every transactions has inputs. Hence, $|\text{Inputs}(\tau)| \geq 1$. Owners of objects $o$ can *express their approval* for transactions $\tau$ that have $o$ as their input. To provide this functionality, we can rely on digital signatures [44].

**Assumption 2.6.** If an owner is well-behaved, then an expression of approval cannot be forged or provided by any other party.[1] Furthermore, a well-behaved owner of $o$ will

---

[1]Earlier, we assumed a unique owner that can approve transactions and prove object ownership in a unique and non-ambiguous way. This does not preclude shared ownership in which multiple participants own an object, however. In that case, we simply require that such a group of participants can approve transactions via their own agreement process to determine which transactions to support (e.g., via multiple signatures, via threshold signatures, or via other mechanisms).

only express its approval for *a single* transaction $\tau$ with $o \in \text{Inputs}(\tau)$, as only one transaction can consume the object $o$, and the owner will only do so after the construction of $o$.

Let $o$ be an object. We assume that there is a well-defined function $\text{shard}(o)$ that maps object $o$ to the single shard $\mathcal{S} \in \text{shards}(\mathfrak{R})$ that is responsible for maintaining $o$. Given a transaction $\tau$, we write $\text{shards}(\tau) = \{\text{shard}(o) \mid o \in \text{Objects}(\tau)\}$ to denote the shards that are affected by $\tau$. Note that the shards $\text{shards}(\tau)$ affected by transaction $\tau$ are all shards that hold either input or output objects of transaction $\tau$. We say that $\tau$ is a *single-shard transaction* if $|\text{shards}(\tau)| = 1$ and is a *multi-shard transaction* otherwise.

## 3 Multi-Shard Transaction Processing

Before we introduce CERBERUS, we put forward the correctness requirements we want to maintain in a multi-shard transaction system in which each shard is itself a set of replicas operated as a Byzantine fault-tolerant system. We say that a shard $\mathcal{S}$ performs an action if every good replica in $\mathcal{G}(\mathcal{S})$ eventually performs that action. Hence, any processing decision or execution step performed by $\mathcal{S}$ requires the usage of a *consensus protocol* [14, 16, 31, 47, 48] that coordinates the operations of individual replicas in the system, e.g., a Byzantine fault-tolerant system driven by PBFT [16], POE [33], or HOTSTUFF [64], or a crash fault-tolerant system driven by PAXOS [47]. As these systems are fully-replicated, each replica will eventually execute the same *sequence of transactions* and, hence, will observe the same evolution of the data held by the system. This *sequence of transactions* is often referred to as a ledger or a journal and is agreed upon via consensus:

**Definition 3.1.** A *consensus protocol* coordinate decision making among the replicas of a resilient cluster $\mathcal{S}$ by providing a reliable ordered replication of *decisions* (e.g., the decision to execute a given transaction as the $\rho$-th transaction processed by the replicas). To do so, consensus protocols provide the following guarantees:

1. If good replica $\text{R} \in \mathcal{S}$ makes a $\rho$-th decision, then all good replicas $\text{R}' \in \mathcal{S}$ will make a $\rho$-th decision (whenever communication becomes reliable).

2. If good replicas $\text{R}, \text{Q} \in \mathcal{S}$ make $\rho$-th decisions, then they make the same decisions.

3. Whenever a good replica learns that a decision $D$ needs to be made, then it can force that all good replicas eventually decide $D$ (even in the presence of malicious behavior).

After a consensus decision is made, it must be preserved by all good replicas: consensus decisions cannot be reverted upon due to crashes or recovery from crashes.

Many definitions of consensus include a requirement of *non-triviality* instead of the last requirement in Definition 3.1. To simplify presentation, we focus on the usage of consensus for operating services that processes transactions requested by clients. In such services, each consensus decision represents a transaction requested by a client and non-triviality is provided by assuring that any client can get their requests processed. Next, we illustrate how.

*Example* 3.2. Assume that consensus is provided via a primary-backup consensus protocol such as PBFT. These primary-backup protocols operate in *views* and within a single view a single replica, the *primary*, coordinates consensus decisions among all other replicas. To deal with primary failures, these primary-backup consensus protocols perform a *view-change*. During the view-change, the system moves to the *next view*. This next view will be coordinated by a *new* primary (typically the next replica assuming some ordering on replicas). An important part of the view-change is that this new primary will learn about and preserve all consensus decisions made in preceding views.

This view-change mechanism is also used to force primaries to propose specific decisions: first, clients can send their requests $D$ to all good replicas; next, all good replicas can suggest $D$ to the primary; finally, either the primary successfully coordinates a consensus decision on $D$ or all good replicas eventually decide that the primary failed to do so and initiate view-changes to replace the primary (until a view is reached in which a good primary proposes the suggested request $D$).

Let $\tau$ be a transaction processed by a sharded fault-tolerant system. Processing of $\tau$ does not imply execution: the transaction could be invalid (e.g., the owners of affected objects did not express their approval) or the transaction could have inputs that no longer exists.

We say that the system *commits* to transaction $\tau$ if the system decides to apply the modifications prescribed by $\tau$, and we say that the system *aborts* $\tau$ if it decides to not do so (after processing it). Finally, we say that a transaction received by the system is *discarded* if it will never be processed (never results in a commit or abort). Note that only *invalid transactions* (e.g., lacking approval of some of the owners of objects affected by the transaction or requests with message format errors) are discarded and this happens before they are considered for execution by the system. All non-discarded transactions will lead to either a commit or an abort decision.

Using this terminology, we put forward the following requirements for any sharded fault-tolerant system:

R1  *Validity*. The system must only commit or abort valid transactions and discard all invalid transactions. In specific, the system must only commit or abort transaction $\tau$ if, for every input object $o \in \texttt{Inputs}(\tau)$ with a well-behaved owner $\texttt{owner}(o)$, the owner $\texttt{owner}(o)$ approves the transaction.[2]

R2  *Shard-involvement*. The shard $\mathcal{S}$ only processes transaction $\tau$ if $\mathcal{S} \in \texttt{shards}(\tau)$.

R3  *Shard-applicability*. Let $D(\mathcal{S})$ be the set of objects maintained by shard $\mathcal{S}$ at time $t$. The shards $\texttt{shards}(\tau)$ can only commit transaction $\tau$ at time $t$ if $\tau$ consumes only objects that exist at time $t$. Hence, $\texttt{Inputs}(\tau) \subseteq \bigcup \{ D(\mathcal{S}) \mid \mathcal{S} \in \texttt{shards}(\tau) \}$.

R4  *Cross-shard-consistency*. If a good replica R partaking in processing transaction $\tau$ concludes that transaction $\tau$ was committed (aborted), then all good replicas in all shards $\mathcal{S}' \in \texttt{shards}(\tau)$ will eventually reach the same conclusion as replica R.

R5  *Service*. If client $c$ is well-behaved and wants to request a valid transaction $\tau$, then the sharded system will eventually *process* $\langle \tau \rangle_c$. If $\tau$ is shard-applicable, then the sharded system will eventually *execute* $\langle \tau \rangle_c$.

R6  *Confirmation*. If the system processes $\langle \tau \rangle_c$ and $c$ is well-behaved, then $c$ will eventually learn whether $\tau$ is committed or aborted.

The *validity* of transactions is a *local requirement*: whether a transaction $\tau$ is valid can be determined by checking whether all owners of inputs of $\tau$ support that transaction. Typically, ownership is expressed via digital signatures, which can be verified deterministically by any replica in any shard independently. Hence, all replicas in all affected shards will make the same conclusion on whether $\tau$ is valid. Likewise, also shard-involvement is a *local requirement*, as individual shards can determine whether they need to process a given transaction. In the same sense, shard-applicability and cross-shard-consistency are *global* requirements, as assuring these requirements requires coordination between the shards affected by a transaction.

In the above and throughout this paper, we will speak of transaction *processing* whenever we look at the steps the system takes after receiving a request (eventually leading to discarding the request when it is invalid, a commit decision, or an abort decision). We will speak of transaction *execution* to refer to transactions that finished processing with either a commit decision or an abort decision.

## 4   Core-CERBERUS: Simple Yet Efficient Transaction Processing

The core idea of CERBERUS is to minimize the coordination necessary for multi-shard ordering and execution of transactions. To do so, CERBERUS combines the semantics of

---

[2]Determining validity of a transaction can include application-level requirements that should hold in a transaction. If, for example, the objects represent monetary balances, then transactions that produce *more output* than they consume *input* can be considered invalid.

transactions in the object-dataset model with the minimal coordination required to assure shard-applicability and cross-shard consistency. This combination results in the following high-level three-step approach towards processing any transaction $\tau$:

1. *Local inputs.* First, every affected shard $S \in \text{shards}(\tau)$ locally determines whether it has all inputs from $S$ that are necessary to process $\tau$.

2. *Cross-shard exchange.* Then, every affected shard $S$ exchanges these inputs to all other shards in $\text{shards}(\tau)$, thereby pledging to use their local inputs when executing of $\tau$.

3. *Decide outcome.* Finally, every affected shard $S$ decides to commit $\tau$ if all affected shards were able to provide all local inputs necessary for execution of $\tau$.

Next, we describe how these three high-level steps are incorporated by CERBERUS into normal consensus steps at each shards. Let shard $S \in \text{shards}(\mathfrak{R})$ receive client request $\langle \tau \rangle_c$. The good replicas in $S$ will first determine whether $\tau$ is valid and whether $\tau$ can be applicable with respect to those inputs $\text{Inputs}(\tau)$ maintained by shard $S$.

If $\tau$ is not valid or $S \notin \text{shards}(\tau)$, then the good replicas discard $\tau$. Otherwise, if $\tau$ is valid and $S \in \text{shards}(\tau)$, then the good replicas can utilize *consensus* to eventually force (Definition 3.1(3)) the shard $S$ to propose a consensus decision as the $\rho$-th consensus decision, for some consensus round $\rho$, on the message $m(S,\tau)_\rho = (\langle \tau \rangle_c, I(S,\tau), D(S,\tau))$, in which $I(S,\tau) = \{ o \in \text{Inputs}(\tau) \mid S = \text{shard}(o) \}$ is the set of objects maintained by $S$ that are input to $\tau$ and $D(S,\tau) \subseteq I(S,\tau)$ is the set of currently-available inputs at $S$. Transaction $\tau$ can only be applicable with respect to those inputs $\text{Inputs}(\tau)$ maintained by shard $S$ if $I(S,\tau) = D(S,\tau)$. Hence, only if $I(S,\tau) = D(S,\tau)$ will $S$ pledge to use the local inputs $I(S,\tau)$ in the execution of $\tau$.

We use *consensus* during the *local inputs* step as it provides an ordered agreement among sequences of transactions. This ordered agreement is necessary to acquire a consistent results among all replicas in a shard: all replicas of a shard need to process all transactions they process in the same order, as otherwise they cannot agree on which of the inputs of a transaction $\tau$ are available to $\tau$ in the presence of other transactions with the same inputs.

The acceptance of $m(S,\tau)_\rho$ in round $\rho$ by all good replicas completes the *local inputs* step. Next, during processing of $\tau$, the *cross-shard exchange* and *decide outcome* steps are performed. First, the *cross-shard exchange* step. In this step, $S$ broadcasts $m(S,\tau)_\rho$ to all other shards in $\text{shards}(\tau)$. To assure that the broadcast arrives, we rely on a reliable primitive for *cross-shard exchange* that guarantees that only approved-upon values can be exchanged. Recently, such primitives have been formalized as cluster-sending [35, 37]:

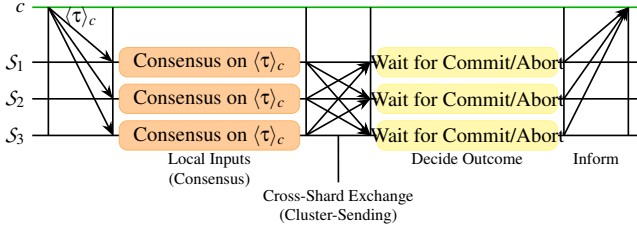

Figure 2: The message flow of CCERBERUS for a 3-shard client request $\langle \tau \rangle_c$ that is committed.

**Definition 4.1.** Let $S_1, S_2$ be two shards. The *cluster-sending problem* is the problem of sending a value $v$ from $S_1$ to $S_2$ such that:

1. all good replicas in $S_2$ *receive* the value $v$;

2. all good replicas in $S_1$ receive *confirmation* that the value $v$ was received by all good replicas in $S_2$; and

3. good replicas in $S_2$ can only receive a value $v$ if all good replicas in $S_1$ *agreed* upon sending $v$.

After $S$ broadcasts $m(S,\tau)_\rho$ to all other shards in $\text{shards}(\tau)$, the replicas in $S$ wait until they receive messages $m(S',\tau)_{\rho'} = (\langle \tau \rangle_c, I(S',\tau), D(S',\tau))$ from all other shards $S' \in \text{shards}(\tau)$.

After cross-shard exchange comes the final *decide outcome* step. After $S$ receives $m(S',\tau)_{\rho'}$ from all shards $S' \in \text{shards}(\tau)$, it decides to *commit* whenever $I(S',\tau) = D(S',\tau)$ for all $S' \in \text{shards}(\tau)$. Otherwise, it decides *abort*. If $S$ decides commit, then all good replicas in $S$ destruct all objects in $D(S,\tau)$ and construct all objects $o \in \text{Outputs}(\tau)$ with $S = \text{shard}(o)$. Finally, each good replica informs $c$ of the outcome of execution. If $c$ receives, from every shard $S'' \in \text{shards}(\tau)$, identical outcomes from $\mathbf{g}_{S''} - \mathbf{f}_{S''}$ distinct replicas in $S''$, then it considers $\tau$ to be successfully executed. In Figure 2, we sketched the working of CCERBERUS.

The *cross-shard exchange* step of CCERBERUS at $S$ involves waiting for other shards $S'$. Hence, there is the danger of deadlocks if the other shards $S'$ never perform their cross-shard exchange steps. To assure that such situations do not lead to a deadlock, we employ two techniques.

1. *Internal propagation.* As shard $S$ has decided *not* to discard $\tau$, the transaction must be valid. Hence, other shards will not discard $\tau$ whenever they receive a request to process $\tau$. To deal with situations in which some shards $S' \in \text{shards}(\tau)$ did not receive $\langle \tau \rangle_c$ (e.g., due to network failure or due to a faulty client that fails to send $\langle \tau \rangle_c$ to $S$), we allow each shard to learn $\tau$ from any other shard. In specific, to assure that all shards $S' \in \text{shards}(\tau)$ will receive a request to process $\tau$, all shards $S'$ will start consensus on $\langle \tau \rangle_c$ after receiving *cross-shard exchange* related to $\langle \tau \rangle_c$. As $S$ uses a resilient primitive

to provide cross-shard exchange, we can assure that all shards $S' \in \text{shards}(\tau)$ will receive cross-shard exchange and eventually start processing $\tau$.

2. *Concurrent resolution.* To deal with concurrent transactions that contend for the same objects, we allow each shard to accept and process transactions for different rounds concurrently. To assure that concurrent resolution does not lead to inconsistent state updates, each replica implements the following *first-pledge* and *ordered-commit* rules. Let $\tau$ be a transaction with $S \in \text{shards}(\tau)$ and $R \in S$. The *first-pledge* rule states that $S$ pledges $o$, constructed in round $\rho$, to transaction $\tau$ only if $\tau$ is the first transaction proposed after round $\rho$ with $o \in \text{Inputs}(\tau)$. The *ordered-commit* rule states that $S$ can abort $\tau$ in any order, but will only commit $\tau$ that is accepted in round $\rho$ after previous rounds finished execution.

The above first-pledge and ordered-commit rules do not need to be enforced or guaranteed, as they specify deterministic behavior for all good replicas. Next, we illustrate the usage of these rules.

*Example* 4.2. Consider two shards $S_1$ and $S_2$ affected by transactions $\tau_1$ and $\tau_2$ that each require objects $o_1$ and $o_2$ residing on shards $S_1$ and $S_2$, respectively. Now consider the case in which shard $S_1$ first processes $\tau_1$ and then $\tau_2$, while shard $S_2$ first processes $\tau_2$ and then $\tau_1$. In this case, shard $S_1$ will pledge $o_1$ to $\tau_1$ and shard $S_2$ will pledge $o_2$ to $\tau_2$. Hence, both $\tau_1$ and $\tau_2$ miss inputs and will fail to complete execution. As both transactions will abort, the order in which they abort does not matter.

In this situation, which will only happen if two transactions have the same inputs in violation of Assumption 2.6, will result in an abort for the two transactions $\tau_1$ and $\tau_2$. Transactions that have unique inputs (in line with Assumption 2.6), will always be able to be committed.

Abort decisions at shard $S$ on a transaction $\tau$ can often be made without waiting for all shards $S' \in \text{shards}(\tau)$: shard $S$ can decide abort after it detects $I(S, \tau) \neq D(S, \tau)$ or after it receives the first message $(\langle \tau \rangle_c, I(S'', \tau), D(S'', \tau))$ with $I(S'', \tau) \neq D(S'', \tau)$, $S'' \in \text{shards}(\tau)$. For efficiency, we allow $S$ to abort in these cases.

**Theorem 4.3.** *If, for all shards $S^*$, $\mathbf{g}_{S^*} > 2\mathbf{f}_{S^*}$, and Assumptions 2.1, 2.5, and 2.6 hold, then Core-*CERBERUS* satisfies Requirements R1–R6 with respect to all transactions that are not requested by malicious clients and do not involve objects with malicious owners.*

*Proof.* Let $\tau$ be a transaction. As good replicas in $S$ discard $\tau$ if it is invalid or if $S \notin \text{shards}(\tau)$, CCERBERUS provides *validity* and *shard-involvement*. Next, *shard-applicability* follow directly from the decide outcome step.

If a shard $S$ commits or aborts transaction $\tau$, then it must have completed the decide outcome and cross-shard exchange steps. Hence, all shards $S' \in \text{shards}(\tau)$ must have exchanged the necessary information to $S$. By relying on cluster-sending for cross-shard exchange, $S'$ requires cooperation of all good replicas in $S'$ to exchange the necessary information to $S$. Hence, we have the guarantee that these good replicas will also perform cross-shard exchange to any other shard $S'' \in \text{shards}(\tau)$. As such, every shard $S'' \in \text{shards}(\tau)$ will receive the same information as $S$, complete cross-shard exchange, and make the same decision during the decide outcome step, providing *cross-shard consistency*.

Due to internal propagation and concurrent resolution, every valid transaction $\tau$ will be processed by CCERBERUS as soon as it is send to any shard $S \in \text{shards}(\tau)$. Hence, every shard in $\text{shards}(\tau)$ will perform the necessary steps to eventually inform the client. As all good replicas $R \in S$, $S \in \text{shards}(\tau)$, will inform the client of the outcome for $\tau$, the majority of these inform-messages come from good replicas, enabling the client to reliably derive the true outcome. Hence, CCERBERUS provides *service* and *confirmation*. $\square$

Notice that in the object-dataset model in which we operate, each object can be constructed once and destructed once. Hence, each object $o$ can be part of at-most two committed transactions: the first of which will construct $o$ as an output, and the second of which has $o$ as an input and will consume and destruct $o$. This is independent of any other operations on other objects. As such these two transactions *cannot* happen concurrently. Consequently, we only have concurrent transactions on $o$ if the owner $\text{owner}(o)$ expresses approval for several transactions that have $o$ as an input. By Assumption 2.6, the owner $\text{owner}(o)$ must be malicious in that case. As such, transactions of well-behaved clients and owners will *never abort*.

In the design of CCERBERUS, we take *full* advantage of the above observation: CCERBERUS effectively *eliminates all coordination* when deciding the order in which individual replicas and shards process both single-shard and multi-shard transactions. Indeed, the order in which replicas will process a multi-shard transaction is decided *before* any coordination and, hence, independent of whichever order the other shards choose to process transactions. Only the outcome of processing transactions are influenced by cross-shard exchange steps. Due to this, CCERBERUS allows all involved shards to process a transaction *independently* with a single consensus step: all communication between shards in CCERBERUS is dedicated to exchange execution state *after* individual shards reach consensus. We can do so as any *aborts*, which could have been prevented with additional coordination, are always due to malicious behavior by clients and owners of objects. Due to this, CCERBERUS will not undo any pledges of objects to the execution of any transactions. This implies that objects that are involved in malicious transactions can get

lost for future usage, while not affecting any transactions of well-behaved clients.

Finally, we remark that CCERBERUS depends on underlying consensus and cluster-sending protocols. The level to which CCERBERUS can deal with asynchronous behavior depends on the particular choices of these protocols.

# 5 Optimistic-CERBERUS: Robust Transaction Processing

In the previous section, we introduced CCERBERUS, a minimalistic multi-shard transaction processing protocol that relies on properties of UTXO-like transactions to maximize performance. Although the design of CCERBERUS is simple yet effective, we see two shortcomings that limits its use. First, CCERBERUS operates under Assumption 2.6, the assumption that any issues arising from concurrent transactions is due to malicious behavior of clients. As such, CCERBERUS chooses to lock out objects affected by such malicious behavior for any future usage. Second, CCERBERUS requires consecutive consensus and cluster-sending steps, which increases its transaction processing latencies. Next, we investigate how to deal with these weaknesses of CCERBERUS *without giving up* on the minimalistic nature of CCERBERUS.

To do so, we propose Optimistic-CERBERUS (OCERBERUS), which is optimized for the *optimistic* case in which we have no concurrent transactions, while providing a recovery path that can recover from concurrent transactions without locking out objects (and without requiring Assumption 2.6). At the core of OCERBERUS is assuring that any issues due to malicious behavior, e.g., concurrent transactions, are *detected* in such a way that individual replicas can recover. At the same time, we want to minimize transaction processing latencies. To bridge between these two objectives, we integrate detection and cross-shard coordination within a single consensus round that runs at each affected shard.

OCERBERUS does not rely on underlying consensus and cluster-sending protocols. For the design of OCERBERUS, we assume *asynchronous communication*: messages can get lost, arrive with arbitrary delays, and in arbitrary order. Consequently, it is impossible to distinguish between, on the one hand, a replica that is malicious and does not send out messages, and, on the other hand, a replica that does send out proposals that get lost in the network. It is well-known that in such an environment, consensus cannot be provided [25, 27]. As such, OCERBERUS is designed to operate in an asynchronous environment in which it will *never cause* data inconsistency and only guarantees *progress* (service and confirmation) eventually when communication is reliable for a sufficiently-long period of time. This is the same model of partial asynchronous communication as used by PBFT.

Assume consensus decisions are made with a PBFT-like primary-backup consensus protocol. Let $\langle\tau\rangle_c$ be a multi-shard transaction, let $\mathcal{S} \in \mathtt{shards}(\tau)$ be an affected shard with primary $\mathcal{P}(\mathcal{S})$, and let $m(\mathcal{S},\tau)_{v,\rho} = (\langle\tau\rangle_c, I(\mathcal{S},\tau), D(\mathcal{S},\tau))$ be the round-$\rho$ proposal of $\mathcal{P}(\mathcal{S})$ of view $v$ of $\mathcal{S}$. To enable detection of concurrent transactions, OCERBERUS modifies the consensus-steps of the underlying consensus protocol by applying the following high-level idea:

> A replica $\mathtt{R} \in \mathcal{S}$, $\mathcal{S} \in \mathtt{shards}(\tau)$, only accepts proposal $m(\mathcal{S},\tau)_{v,\rho}$ for transaction $\tau$ if it gets confirmation that replicas in each other shard $\mathcal{S}' \in \mathtt{shards}(\tau)$ are also accepting proposals for $\tau$. Otherwise, replica $\mathtt{R}$ detects failure.

To simplify presentation, we will use a traditional design that uses all-to-all communication between all replicas in all affected shards akin to the design of PBFT [16]. Traditionally, PBFT has a *commit* phase that is distinct from the commit decision made during multi-shard transaction processing. To disambiguate the commit phase of PBFT and the commit decision made during multi-shard transaction processing, we have renamed the commit phase of PBFT into the *finalize* phase. To minimize inter-shard communication (at the cost of latency) one can also utilize threshold signatures to implement all-to-one-to-all communication akin to the design of HOTSTUFF [64] to carry over local prepare and finalize certificates between shards via a few constant-sized messages.[3]

Next, we illustrate how to integrate the above idea in the three-phase design of PBFT, thereby turning PBFT into a multi-shard aware consensus protocol:

1. *Global preprepare.* Primary $\mathcal{P}(\mathcal{S})$ must send $m(\mathcal{S},\tau)_{v,\rho}$ to all replicas $\mathtt{R}' \in \mathcal{S}'$, $\mathcal{S}' \in \mathtt{shards}(\tau)$. Replica $\mathtt{R} \in \mathcal{S}$ only finishes the global preprepare phase after it receives a *global preprepare certificate* consisting of a set $M = \{m(\mathcal{S}'',\tau)_{v'',\rho''} \mid \mathcal{S}'' \in \mathtt{shards}(\tau)\}$ of preprepare messages from all primaries of shards affected by $\tau$.

2. *Global prepare.* After $\mathtt{R} \in \mathcal{S}$, $\mathcal{S} \in \mathtt{shards}(\tau)$, finishes the global preprepare phase, it sends prepare messages for $M$ to all other replicas in $\mathtt{R}' \in \mathcal{S}'$, $\mathcal{S}' \in \mathtt{shards}(\tau)$. Replica $\mathtt{R} \in \mathcal{S}$ only finishes the global prepare phase for $M$ after, for every shard $\mathcal{S}' \in \mathtt{shards}(\tau)$, it receives a *local prepare certificate* consisting of a set $P(\mathcal{S}')$ of prepare messages for $M$ from $\mathbf{g}_{\mathcal{S}'}$ distinct replicas in $\mathcal{S}'$. We call the set $\{P(\mathcal{S}'') \mid \mathcal{S}'' \in \mathtt{shards}(\tau)\}$ a *global prepare certificate*.

---

[3]Such a design will require a single replica responsible for receiving all messages, aggregating them into a single certificate, and broadcasting this message back. To deal with failures of this replica, we can simply choose a primary (whose failures are already dealt with by the normal recovery mechanisms that we shall describe). Alternatively, one can use the approach taken by SBFT [28] and choose any other replica as the aggregator (thereby offloading the primary), but this will require additional recovery mechanisms to deal with failures of that replica.

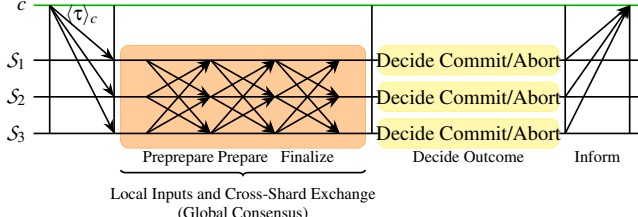

Figure 3: The message flow of OCERBERUS for a 3-shard client request $\langle\tau\rangle_c$ that is committed.

3. *Global finalize.* After replica $R \in S$, $S \in \text{shards}(\tau)$, finishes the global prepare phase, it sends finalize messages for $M$ to all other replicas in $R' \in S'$, $S' \in \text{shards}(\tau)$. Replica $R \in S$ only finishes the global finalize phase for $M$ after, for every shard $S' \in \text{shards}(\tau)$, it receives a *local finalize certificate* consisting of a set $C(S')$ of finalize messages for $M$ from $\mathbf{g}_{S'}$ distinct replicas in $S'$. We call the set $\{P(S'') \mid S'' \in \text{shards}(\tau)\}$ a *global finalize certificate*.

The above three-phase *global*-PBFT protocol already takes care of the *local input* and *cross-shard exchange* steps. Indeed, a replica $R \in S$ that finishes the global finalize phase has accepted global preprepare certificate $M$, which contains all information of other shards to proceed with processing. At the same time, $R$ also has confirmation that $M$ is prepared by a majority of all good replicas in each shard $S' \in \text{shards}(\tau)$ (which will eventually be followed by execution of $\tau$ within $S'$). With these ingredients in place, only the *decide outcome* step remains.

The decide outcome step at shard $S$ is entirely determined by the global preprepare certificate $M$. Shard $S$ decides to *commit* whenever $I(S', \tau) = D(S', \tau)$ for all $(\langle\tau\rangle_c, I(S', \tau), D(S', \tau)) \in M$. Otherwise, it decides *abort*. If $S$ decides commit, then all good replicas in $S$ destruct all objects in $D(S, \tau)$ and construct all objects $o \in \text{Outputs}(\tau)$ with $S = \text{shard}(o)$. Finally, each good replica informs $c$ of the outcome of execution. If $c$ receives, from every shard $S' \in \text{shards}(\tau)$, identical outcomes from $\mathbf{g}_{S'} - \mathbf{f}_{S'}$ distinct replicas in $S'$, then it considers $\tau$ to be successfully executed. In Figure 3, we sketched the working of OCERBERUS.

We note that the multi-shard aware consensus protocol OCERBERUS is a *multi-shard consensus protocol* that aims to make a single consensus decision among all replicas in all shards affected by a multi-shard transaction. OCERBERUS is not the only multi-shard consensus protocol recently proposed (e.g., [3, 4]). What sets OCERBERUS apart is how it guarantees correctness *in all environments*, which is determined by how OCERBERUS deals with *non-optimistic cases* in which failure is detected and recovery is necessary. We will detail recovery next. As a first step, we illustrate the ways in which the normal-case of OCERBERUS can fail (e.g., due

to malicious behavior of clients, failing replicas, or unreliable communication).

*Example* 5.1. Consider a transaction $\tau$ proposed by client $c$ and affecting shard $S \in \text{shards}(\tau)$. First, we consider the case in which $\mathcal{P}(S)$ is malicious and tries to set up a coordinated attack. To have maximum control over the steps of OCERBERUS, the primary sends the message $m(S, \tau)_{v,\rho}$ to only $\mathbf{g}_{S''} - \mathbf{f}_{S''}$ good replicas in each shard $S'' \in \text{shards}(\tau)$. By doing so, $\mathcal{P}(S)$ can coordinate the faulty replicas in each shard to assure failure of any phase at any replica $R' \in S'$, $S' \in \tau$:

1. To prevent $R'$ from finishing the global preprepare phase (and start the global prepare phase) for an $M$ with $m(S', \tau)_{v', \rho'} \in M$, $\mathcal{P}(S)$ simply does not send $m(S, \tau)_{v, \rho}$ to $R'$.

2. To prevent $R'$ from finishing the global prepare phase (and start the global finalize phase) for $M$, $\mathcal{P}(S)$ instructs the faulty replicas in $\mathcal{F}(S)$ to not send prepare messages for $M$ to $R'$. Hence, $R'$ will receive at-most $\mathbf{g}_S - \mathbf{f}_S$ prepare messages for $M$ from replicas in $S$, assuring that it will not receive a local prepare certificate $P(S)$ and will not finish the global prepare phase for $M$.

3. Likewise, to prevent $R'$ from finishing the global finalize phase (and start execution) for $M$, $\mathcal{P}(S)$ instructs the faulty replicas in $\mathcal{F}(S)$ to not send finalize messages to $R'$. Hence, $R'$ will receive at-most $\mathbf{g}_S - \mathbf{f}_S$ finalize messages for $M$ from replicas in $S$, assuring that it will not receive a local finalize certificate $C(S)$ and will not finish the global finalize phase for $M$.

None of the above attacks can be attributed to faulty behavior of $\mathcal{P}(S)$ as unreliable communication can result in the same outcomes for $R'$. Furthermore, even if communication is reliable and $\mathcal{P}(S)$ is good, replica $R'$ can see the same outcomes due to malicious behavior of the client or of primaries of other shards in $\text{shards}(\tau)$:

1. The client $c$ can be malicious and not send $\tau$ to $S$. At the same time, all other primaries $\mathcal{P}(S'')$ of shards $S'' \in \text{shards}(\tau)$ can be malicious and not send anything to $S$ either. In this case, $\mathcal{P}(S)$ will never be able to send any message $m(S, \tau)_{v,\rho}$ to $R'$, as no replica in $S$ is aware of $\tau$.

2. If any primary $\mathcal{P}(S'')$ of $S'' \in \text{shards}(\tau)$ is malicious, then it can prevent some replicas in $S$ from starting the global prepare phase, thereby preventing these replicas to send prepare messages to $R'$. If $\mathcal{P}(S'')$ prevents sufficient replicas in $S$ from starting the global prepare phase, $R'$ will be unable to finish the global prepare phase.

3. Likewise, any malicious primary $\mathcal{P}(S'')$ of $S'' \in \text{shards}(\tau)$ can prevent replicas in $S$ from starting the

global finalize phase, thereby assuring that R′ will be unable to finish the global finalize phase.

To deal with malicious behavior, OCERBERUS needs a robust recovery mechanism. Indeed, the main difference of the multi-shard consensus of OCERBERUS and the single-shard consensus PBFT is that OCERBERUS will use a single primary *per* shard whereas PBFT only has a single primary. This difference affects the capability of individual replicas to detect the root cause of disruptions of the normal-case operations (as several primaries could be the root cause of such disruptions). As such, we cannot simply build the robust recovery mechanism on top of traditional view-change approaches: these traditional view-change approaches require that one can identify a single source of failure (when communication is reliable), namely the current primary. To remedy this, the recovery mechanisms of OCERBERUS has components that perform *local view-change* and that perform *global state recovery*.

Next, we will detail the working of the recovery mechanisms of OCERBERUS. To simplify presentation, we will focus on the recovery of a single transaction. The techniques presented are straightforward to generalize to any history of zero-or-more transactions. The pseudo-code for the recovery protocol can be found in Figure 4. Next, we describe the working of this recovery protocol in detail.

Let $R \in \mathcal{S}$ be a replica that determines that it cannot finish a round $\rho$ of view $v$. First, R determines whether it already has a *guarantee* on which transaction it has to process in round $\rho$. This is the case when the following conditions are met: R finished the global prepare phase for $M$ with $m(\mathcal{S}, \tau)_{v,\rho} \in M$ and has received a local finalize certificate $C(\mathcal{S}'')$ for $M$ from some shard $\mathcal{S}'' \in \text{shards}(\tau)$. In this case, R can simply request all missing local finalize certificates directly, as $C(\mathcal{S}'')$ can be used to prove to any involved replica $R' \in \mathcal{S}'$, $\mathcal{S}' \in \text{shards}(\tau)$, that R′ also needs to finalize $M$. To request such missing finalize certificates of $\mathcal{S}'$, replica R sends out VCGlobalSCR messages to all replicas in $\mathcal{S}'$ (Line 7 of Figure 4). Any replica R′ that receives such a VCGlobalSCR message can use the information in that message to reach the global finalize phase for $M$ and, hence, provide R with the requested finalize messages (Line 11 of Figure 4).

If R does not have a *guarantee* itself on which transaction it has to process in round $\rho$, then it needs to determine whether any other replica (either in its own shard or in any other shard) has already received and acted upon such a guarantee. To initiate such local and global state recovery, R simply detects the current view as faulty. To do so, R broadcasts a VCRecoveryRQ message to all other replicas in $\mathcal{S}$ that contains all information R collected on round $\rho$ in view $v$ (Line 4 of Figure 5). Other replicas $Q \in \mathcal{S}$ that already have *guarantees* for round $\rho$ can help R by providing all missing information (Line 6 of Figure 5). On receipt of this information, R can proceed with the round (Line 7 of Figure 5). If no replicas can provide the missing information, then eventually

```
 1: event R ∈ 𝒮 is unable to finish round ρ of view v do
 2:     if R finished in round ρ the global prepare phase for M,
            but is unable to finish the global finalize phase then
 3:         Let P be the global prepare certificate of R for M.
 4:         if R has a local finalize certificate C(𝒮″) for M then
 5:             for 𝒮′ ∈ shards(τ) do
 6:                 if R did not yet receive a local finalize certificate C(𝒮′) then
 7:                     Broadcast ⟨VCGlobalSCR : M, P, C(𝒮″)⟩ to all replicas in 𝒮′.
 8:             else Detect the need for local state recovery of round ρ of view v (Figure 5).
 9:         else Detect the need for local state recovery of round ρ of view v (Figure 5).
10:         (Eventually repeat this event if R remains unable to finish round ρ.)

11: event R′ ∈ 𝒮′ receives message ⟨VCGlobalSCR : M, P, C(𝒮″)⟩ from R ∈ 𝒮 do
12:     if R′ did not reach the global finalize phase for M then
13:         Use M, P, and C(𝒮″) to reach the global finalize phase for M.
14:     else Send a finalize message for M to R.
```

Figure 4: The view-change *global short-cut recovery path* that determines whether R already has the assurance that the current transaction will be finalized. If this is the case, then R requests only the missing information to proceed with execution. Otherwise, R requires at-least local recovery (Figure 5).

all good replicas will detect the need for local recovery, this either by themselves (Line 1 of Figure 5) or after receiving VCRecoveryRQ messages of at-least $\mathbf{f}_\mathcal{S} + 1$ distinct replicas in $\mathcal{S}$, of which at-least a single replica must be good (Line 10 of Figure 5).

Finally, if a replica R receives $\mathbf{g}_\mathcal{S}$ VCRecoveryRQ messages, then it has the guarantee that at least $\mathbf{g}_\mathcal{S} - \mathbf{f}_\mathcal{S} \geq \mathbf{f}_\mathcal{S} + 1$ of these messages come from good replicas in $\mathcal{S}$. Hence, due to Line 10 of Figure 5, all $\mathbf{g}_\mathcal{S}$ good replicas in $\mathcal{S}$ will send VCRecoveryRQ, and, when communication is reliable, also receive these messages. Consequently, at this point, R can start the new view by electing a new primary and awaiting the NewView proposal of this new primary (Line 12 of Figure 5). If R is the new primary, then it starts the new view by proposing a NewView. As other shards *could* have already made final decisions depending on local prepare or finalize certificates of $\mathcal{S}$ for round $\rho$, we need to assure that such certificates are not invalidated. To figure out whether such final decisions have been made, the new primary will query other shards $\mathcal{S}'$ for their state whenever the NewView message contains global preprepare certificates for transactions $\tau$, $\mathcal{S}' \in \text{shards}(\tau)$, but not a local finalize certificate to *guarantee* execution of $\tau$ (Line 17 of Figure 5).

The new-view process has three stages. First, the new primary P proposes the new-view via a NewView message (Line 12 of Figure 5). If necessary, the new primary P also requests the relevant global state from any relevant shard (Line 1 of Figure 6). The replicas in other shards will respond to this request with their local state (Line 9 of Figure 6). The new primary collects these responses and sends them to all replicas in $\mathcal{S}$ via a NewViewGlobal message. Then, after P sends the NewView message to $R \in \mathcal{S}$, R determines whether the NewView message contains sufficient information to recover round $\rho$ (Line 15 of Figure 6), contains sufficient information to wait for any relevant global state (Line 17 of

```
1:  event R ∈ S detects the need for local state recovery of round ρ of view v do
2:      Let M be any latest global preprepare certificate accepted for round ρ by R.
3:      Let S be M and any prepare and finalize certificates for M collected by R.
4:      Broadcast ⟨VCRecoveryRQ : v,ρ,S⟩.

5:  event Q ∈ S receives messages ⟨VCRecoveryRQ : v,ρ,S⟩ of R ∈ S and Q has
        1. started the global prepare phase for M with m(S,τ)_{w,ρ} ∈ M;
        2. a global prepare certificate for M;
        3. a local finalize certificate C(S″) for M
    do
6:      Send ⟨VCLocalSCR : M,P,C(S″)⟩ to R ∈ S.

7:  event R ∈ S receives message ⟨VCLocalSCR : M,P,C(S″)⟩ from Q ∈ S do
8:      if R did not reach the global finalize phase for M then
9:          Use M, P, and C to reach the global finalize phase for M.

10: event R ∈ S receives messages ⟨VCRecoveryRQ : v_i,ρ,S_i⟩, 1 ≤ i ≤ f_S + 1,
        from f_S + 1 distinct replicas in S do
11:     R detects the need for local state recovery of round ρ
        of view min{v_i | 1 ≤ i ≤ f_S + 1}.

12: event R ∈ S receives messages ⟨VCRecoveryRQ : v,ρ,S_i⟩, 1 ≤ i ≤ g_S,
        from distinct replicas in S do
13:     if id(R) ≠ (v+1) mod n_S then
14:         (R awaits the NewView message of the new primary, Line 14 of Figure 6.)
15:     else
16:         Broadcast ⟨NewView : ⟨VCRecoveryRQ : v,ρ,S_i⟩ | 1 ≤ i ≤ g_S⟩ to
            all replicas in S.
17:         if there exists a S_i that contains global preprepare certificate M,
                but no S_j contains a local finalize certificate for M then
18:             R initiates global state recovery of round ρ (Line 1 of Figure 6).
```

Figure 5: The view-change *local short-cut recovery path* that determines whether some Q can provide R with the assurance that the current transaction will be finalized. If this is the case, then R only needs this assurance, otherwise S requires a new view (Figure 6).

```
1:  event P ∈ S initiates global state recovery of round ρ using ⟨NewView : V⟩ do
2:      Let T be the transactions with global preprepare certificates for round ρ of S in
        view V.
3:      Let S be the shards affected by transactions in T.
4:      Broadcast ⟨VCGlobalStateRQ : v,ρ,V⟩ to all replicas in S′ ∈ S.
5:      for S′ ∈ S do
6:          Wait for VCGlobalStateRQ messages for V from g_{S′} distinct replicas in S′.
7:          Let W(S′) be the set of received VCGLOBALSTATERQ messages.
8:      Broadcast ⟨NewViewGlobal : V,{W(S′) | S′ ∈ S}⟩
        to all replicas in S.

9:  event R′ ∈ S′ receives message ⟨VCGlobalStateRQ : v,ρ,V⟩ from P ∈ S do
10:     if R′ has a global preprepare certificate M with m(S,τ)_{w,ρ} ∈ M
            and reached the global finalize phase for M then
11:         Let P be the global prepare certificate for M.
12:         Send ⟨VCGlobalStateR : v,ρ,V,M,P⟩ to P.
13:     else Send ⟨VCGlobalStateR : v,ρ,V⟩ to P.

14: event R ∈ S receives valid ⟨NewView : V⟩ message from replica P do
15:     if there exists a ⟨VCRecoveryRQ : v_i,ρ,S_i⟩ ∈ V that contains
            1. a global preprepare certificate M with m(S,τ)_{w,ρ} ∈ M;
            2. a global prepare certificate P for M; and
            3. a local finalize certificate C(S″) for M
        then
16:         Use M, P, and C to reach the global finalize phase for M.
17:     else if there exists a ⟨VCRecoveryRQ : v_i,ρ,S_i⟩ ∈ V that contains
            1. a global preprepare certificate M; and
            2. no ⟨VCRecoveryRQ : v_j,ρ,S_j⟩ ∈ V contains a
               local finalize certificate for M
        then
18:         R detects the need for global state recovery of round ρ (Line 20 of Figure 6).
19:     else (P must propose for round ρ.)

20: event R ∈ S receives valid ⟨NewViewGlobal : V,W⟩ from P ∈ S do
21:     if any message in W is of the form ⟨VCGlobalStateR : v,ρ,V,M,P⟩ then
22:         Select ⟨VCGlobalStateR : v,ρ,V,M,P⟩ ∈ W with
            latest view w, m(S,τ)_{w,ρ} ∈ M.
23:         Use M and P to reach the global finalize phase for M.
24:     else (P must propose for round ρ.)
```

Figure 6: The view-change *new-view recovery path* that recovers the state of the previous view based on a NewView proposal of the new primary. As part of the new-view recovery path, the new primary can construct a global new-view that contains the necessary information from other shards to reconstruct the local state.

Figure 6), or to determine that the new primary must propose for round ρ (Line 19 of Figure 6). If R determines it needs to wait for any relevant global state, then R will wait for this state to arrive via a NewViewGlobal message. Based on the received global state, R determines to recover round ρ (Line 21 of Figure 6), or determines that the new primary must propose for round ρ (Line 24 of Figure 6).

Next, we will prove the correctness of the view-change of OCERBERUS. First, using a standard quorum argument, we prove that in a single round of a single view of S, only a single global preprepare message affecting S can get finalized by any other affected shards:

**Lemma 5.1.** *Let $\tau_1$ and $\tau_2$ be transactions with $S \in (shards(\tau_1) \cap shards(\tau_2))$. If $\mathbf{g}_S > 2\mathbf{f}_S$ and there exists shards $S_i \in shards(\tau_i)$, $i \in \{1,2\}$, such that good replicas $R_i \in \mathcal{G}(S_i)$ reached the global finalize phase for global preprepare certificate $M_i$ with $m(S,\tau_i)_{v,\rho} \in M_i$, then $\tau_1 = \tau_2$.*

*Proof.* We prove this property using contradiction. We assume $\tau_1 \neq \tau_2$. Let $P_i(S)$ be the local prepare certificate provided by $S$ for $M_i$ and used by $R_i$ to reach the global finalize phase, let $S_i \subseteq S$ be the $\mathbf{g}_S$ replicas in $S$ that provided the prepare messages in $P_i(S)$, and let $T_i = S_i \setminus \mathcal{F}(S)$ be the good replicas in $S_i$. By construction, we have $|T_i| \geq \mathbf{g}_S - \mathbf{f}_S$. As all replicas in $T_1 \cup T_2$ are good, they will only send out a single prepare message per round ρ of view $v$. Hence, if $\tau_1 \neq \tau_2$,

then $T_1 \cap T_2 = \emptyset$, and we must have $2(\mathbf{g}_S - \mathbf{f}_S) \leq |T_1 \cup T_2|$. As all replicas in $T_1 \cup T_2$ are good, we also have $|T_1 \cup T_2| \leq \mathbf{g}_S$. Hence, $2(\mathbf{g}_S - \mathbf{f}_S) \leq \mathbf{g}_S$, which simplifies to $\mathbf{g}_S \leq 2\mathbf{f}_S$, a contradiction. Hence, we conclude $\tau_1 = \tau_2$. □

Next, we use Lemma 5.1 to prove that any global prepare certificate that *could* have been accepted by any good affected replica is preserved by OCERBERUS:

**Proposition 5.1.** *Let $\tau$ be a transaction and $m(S,\tau)_{v,\rho}$ be a preprepare message. If, for all shards $S^*$, $\mathbf{g}_{S^*} > 2\mathbf{f}_{S^*}$, and there exists a shard $S' \in shards(\tau)$ such that $\mathbf{g}_{S'} - \mathbf{f}_{S'}$ good replicas in $S'$ reached the global finalize phase for $M$ with $m(S,\tau)_{v,\rho} \in M$, then every successful future view of $S$ will recover $M$ and assure that the good replicas in $S$ reach the finalize phase for $M$.*

*Proof.* Let $v^* \leq v$ be the first view in which a global prepare certificate $M^*$ with $m(S,\tau^*)_{v^*,\rho} \in M^*$ satisfied the premise of this proposition. Using induction on the number of views after

the first view $v^*$, we will prove the following two properties on $M^*$:

1. every good replica that participates in view $w$, $v^* < w$, will recover $M^*$ upon entering view $w$ and reach the finalize phase for $M^*$; and

2. no replica will be able to construct a local prepare certificate of $\mathcal{S}$ for any global preprepare certificate $M^\dagger \neq M^*$ with $m(\mathcal{S}, \tau^\dagger)_{w,\rho} \in M^\dagger$, $v^* < w$.

The base case is view $v^* + 1$. Let $S' \subseteq \mathcal{G}(\mathcal{S}')$ be the set of $\mathbf{g}_{\mathcal{S}'} - \mathbf{f}_{\mathcal{S}'}$ good replicas in $\mathcal{S}'$ that reached the global finalize phase for $M^*$. Each replica $\mathrm{R}' \in S'$ has a local prepare certificate $P(\mathcal{S})$ consisting of $\mathbf{g}_{\mathcal{S}}$ prepare messages for $M^*$ provided by replicas in $\mathcal{S}$. We write $S(\mathrm{R}') \subseteq \mathcal{G}(\mathcal{S})$ to denote the at-least $\mathbf{g}_{\mathcal{S}} - \mathbf{f}_{\mathcal{S}}$ good replicas in $\mathcal{S}$ that provided such a prepare message to $\mathrm{R}'$.

Consider any valid new-view proposal $\langle \texttt{NewView} : V \rangle$ for view $v^* + 1$. If the conditions of Line 15 of Figure 6 hold for global preprepare certificate $M^\dagger$ with $m(\mathcal{S}, \tau^\ddagger)_{w,\rho} \in M^\ddagger$, then we recover $M^\ddagger$. As there is a local finalize certificate for $M^\ddagger$ in this case, the premise of this proposition holds on $M^\ddagger$. As $v^*$ is the first view in which the premise of this proposition hold, we can use Lemma 5.1 to conclude that $w = v^*$, $M^\ddagger = M^*$, and, hence, that the base case holds if the conditions of Line 15 of Figure 6 hold. Next, we assume that the conditions of Line 15 of Figure 6 do not hold, in which case $M^*$ can only be recovered via global state recovery. As the first step in global state recovery is proving that the condition of Line 17 of Figure 6 holds. Let $T \subseteq \mathcal{G}(\mathcal{S})$ be the set of at-least $\mathbf{g}_{\mathcal{S}} - \mathbf{f}_{\mathcal{S}}$ good replicas in $\mathcal{S}$ whose $\texttt{VCRecoveryRQ}$ message is in $V$ and let $\mathrm{R}' \in S'$. We have $|S(\mathrm{R}')| \geq \mathbf{g}_{\mathcal{S}} - \mathbf{f}_{\mathcal{S}}$ and $|T| \geq \mathbf{g}_{\mathcal{S}} - \mathbf{f}_{\mathcal{S}}$. Hence, by a standard quorum argument, we conclude $S(\mathrm{R}') \cap T \neq \emptyset$. Let $\mathrm{Q} \in (S(\mathrm{R}') \cap T)$. As $\mathrm{Q}$ is good and send prepare messages for $M^*$, it must have reached the global prepare phase for $M^*$. Consequently, the condition of Line 17 of Figure 6 holds and to complete the proof, we only need to prove that any well-formed $\texttt{NewViewGlobal}$ message will recover $M^*$.

Let $\langle \texttt{NewViewGlobal} : V, W \rangle$ be any valid global new-view proposal for view $v^* + 1$. As $\mathrm{Q}$ reached the global prepare phase for $M^*$, any valid global new-view proposal must include messages from $\mathcal{S}' \in \texttt{shards}(\tau)$. Let $U' \subseteq \mathcal{S}'$ be the replicas in $\mathcal{S}'$ of whom messages $\texttt{VCGlobalStateR}$ are included in $W$. Let $V' = U' \setminus \mathcal{F}(\mathcal{S}')$. We have $|S'| \geq \mathbf{g}_{\mathcal{S}'} - \mathbf{f}_{\mathcal{S}'}$ and $|V'| \geq \mathbf{g}_{\mathcal{S}'} - \mathbf{f}_{\mathcal{S}'}$. Hence, by a standard quorum argument, we conclude $S' \cap V' \neq \emptyset$. Let $\mathrm{Q}' \in (S' \cap V')$. As $\mathrm{Q}'$ reached the global finalize phase for $M^*$, it will meet the conditions of Line 23 of Figure 6 and provide both $M^*$ and a global prepare certificate for $M^*$. Let $M^\ddagger$ be any other global preprepare certificate in $W$ accompanied by a global prepare certificate. Due to Line 22 of Figure 6, the global preprepare certificate for the newest view of $\mathcal{S}$ will be recovered. As $v^*$ is the newest view of $\mathcal{S}$, $M^\ddagger$ will only prevent recovery of $M^*$ if it is also

a global preprepare certificate for view $v^*$ of $\mathcal{S}$. In this case, Lemma 5.1 guarantees that $M^\ddagger = M^*$. Hence, any replica $\mathrm{R}$ will recover $M^*$ upon receiving $\langle \texttt{NewViewGlobal} : V, W \rangle$.

Now assume that the induction hypothesis holds for all views $j$, $v^* < j \leq i$. We will prove that the induction hypothesis holds for view $i + 1$. Consider any valid new-view proposal $\langle \texttt{NewView} : V \rangle$ for view $i + 1$ and let $M^\ddagger$ with $m(\mathcal{S}, \tau^\ddagger)_{w,\rho} \in M^\ddagger$ be any global preprepare certificate that is recovered due to the new-view proposal $\langle \texttt{NewView} : V \rangle$. Hence, $M^\ddagger$ is recovered via either Line 16 of Figure 6 or Line 23 of Figure 6. In both cases, there must exist a global prepare certificate $P$ for $M^\ddagger$. As $\langle \texttt{NewView} : V \rangle$ is valid, we must have $w \leq i$. Hence, we can apply the second property of the induction hypothesis to conclude that $w \leq v^*$. If $w = v^*$, then we can use Lemma 5.1 to conclude that $M^\ddagger = M^*$. Hence, to complete the proof, we must show that $w = v^*$. First, the case in which $M^\ddagger$ is recovered via Line 16 of Figure 6. Due to the existence of a global finalize certificate $C$ for $M^\ddagger$, $M^\ddagger$ satisfies the premise of this proposition. By assumption, $v^*$ is the first view for which the premise of this proposition holds. Hence, $w \geq v^*$, in which case we conclude $M^\ddagger = M^*$. Last, the case in which $M^\ddagger$ is recovered via Line 23 of Figure 6. In this case, $M^\ddagger$ is recovered via some message $\langle \texttt{NewViewGlobal} : V, W \rangle$. Analogous to the proof for the base case, $V$ will contain a message $\texttt{VCRecoveryRQ}$ from some replica $\mathrm{Q} \in S(\mathrm{R}')$. Due to Line 2 of Figure 5, $\mathrm{Q}$ will provide information on $M^*$. Consequently, a prepare certificate for $M^*$ will be obtained via global state recovery, and we also conclude $M^\ddagger = M^*$. $\qquad\square$

Lemma 5.1 and Proposition 5.1 assure that no transaction that could-be-finalized by any replica will ever get lost by the system. Next, we bootstrap these technical properties to prove that all good replicas can always recover such could-be-finalized transactions.

**Proposition 5.2.** *Let $\tau$ be a transaction and $m(\mathcal{S}, \tau)_{v,\rho}$ be a preprepare message. If, for all shards $\mathcal{S}^*$, $\mathbf{g}_{\mathcal{S}^*} > 2\mathbf{f}_{\mathcal{S}^*}$, and there exists a shard $\mathcal{S}' \in \texttt{shards}(\tau)$ such that $\mathbf{g}_{\mathcal{S}'} - \mathbf{f}_{\mathcal{S}'}$ good replicas in $\mathcal{S}'$ reached the global finalize phase for $M$ with $m(\mathcal{S}, \tau)_{v,\rho} \in M$, then every good replica in $\mathcal{S}$ will accept $M$ whenever communication becomes reliable.*

*Proof.* Let $\mathrm{R} \in \mathcal{S}$ be a good replica that is unable to accept $M$. At some point, communication becomes reliable, after which $\mathrm{R}$ will eventually trigger Line 1 of Figure 4. We have the following cases:

1. If $\mathrm{R}$ meets the conditions of Line 4 of Figure 4, then $\mathrm{R}$ has a local finalize certificate $C(\mathcal{S}'')$, $\mathcal{S}'' \in \texttt{shards}(\tau)$. This local finalize certificate certifies that at least $\mathbf{g}_{\mathcal{S}''} - \mathbf{f}_{\mathcal{S}''}$ good replicas in $\mathcal{S}''$ finished the global prepare phase for $M$. Hence, the conditions for Proposition 5.1 are met for $M$ and, hence, any shard in $\texttt{shards}(\tau)$ will maintain or recover $M$. Replica $\mathrm{R}$ can use $C(\mathcal{S}'')$ to prove this situation to other replicas, forcing them to finalize $M$,

and provide any finalize messages R is missing (Line 11 of Figure 4).

2. If R does not meet the conditions of Line 4 of Figure 4, but some other good replica Q ∈ $\mathcal{S}$ does, then Q can provide all missing information to R (Line 6 of Figure 5). Next, R uses this information (Line 7 of Figure 5), after which it meets the conditions of Line 4 of Figure 4.

3. Otherwise, if the above two cases do not hold, then all $\mathbf{g}_{\mathcal{S}}$ good replicas in $\mathcal{S}$ are unable to finish the finalize phase. Hence, they perform a view-change. Due to Proposition 5.1, this view-change will succeed and put every replica in $\mathcal{S}$ into the finalize phase for $M$. As all good replicas in $\mathcal{S}$ are in the finalize phase, each good replica in $\mathcal{S}$ will be able to make a local finalize certificate $C(\mathcal{S})$ for $M$, after which they meet the conditions of Line 4 of Figure 4.                                                                                  □

Finally, we use Proposition 5.2 to prove *cross-shard-consistency*.

**Theorem 5.2.** *Optimistic*-CERBERUS *maintains cross-shard consistency.*

*Proof.* Assume a single good replica R ∈ $\mathcal{S}$ executes a transaction $\tau$ (by committing or aborting). Hence, it accepted some global preprepare certificate $M$ with $m(\mathcal{S},\tau)_{v,\rho} \in M$. Consequently, R has local finalize certificates $C(\mathcal{S}')$ for $M$ of every $\mathcal{S}' \in \text{shards}(\tau)$. Hence, at least $\mathbf{g}_{\mathcal{S}'} - \mathbf{f}_{\mathcal{S}'}$ good replicas in $\mathcal{S}'$ reached the global finalize phase for $M$, and we can apply Proposition 5.2 to conclude that any good replica R″ ∈ $\mathcal{S}''$, $\mathcal{S}'' \in \text{shards}(\tau)$ will accept $M$. As R″ bases its execution decision for $\tau$ on the same global prepare certificate $M$ as R, they will both make the same decision, completing the proof.                                                                                  □

Due to the similarity between OCERBERUS and CCERBERUS, one can use the details of Theorem 4.3 to prove that OCERBERUS provides *validity*, *shard-involvement*, and *shard-applicability*. Via Theorem 5.2, we proved *cross-shard-consistency*. We cannot prove *service* and *confirmation*, however. The reason for this is simple: even though OCERBERUS can detect and recover from accidental faulty behavior and accidental concurrent transactions, OCERBERUS is not designed to gracefully handle targeted attacks: OCERBERUS is optimistic in the sense that it is optimized for the situation in which faulty behavior (including concurrent transactions that contend for the same objects) is rare. Still, in all cases, OCERBERUS maintains cross-shard consistency, however. Moreover, in the optimistic case, progress is guaranteed:

**Proposition 5.3.** *If, for all shards $\mathcal{S}^*$, $\mathbf{g}_{\mathcal{S}^*} > 2\mathbf{f}_{\mathcal{S}^*}$, and Assumptions 2.1, 2.5, and 2.6 hold, then Optimistic-*CERBERUS *satisfies Requirements R1–R6 in the optimistic case (whenever communication is reliable, shards have good primaries, and no concurrent transactions exist).*

OCERBERUS cannot defend against denial-of-service attacks targeted at blocking individual replicas and shards from participating. Unfortunately, no existing consensus protocol is able to deal with such attacks. Furthermore, as is the case for other multi-shard consensus protocols, coordinated attempts can prevent OCERBERUS from making progress in periods when the optimistic assumption does not hold. At the core of such attacks is the ability for malicious clients and malicious primaries to corrupt the operations of shards coordinated by good primaries, as already shown in Example 5.1. Due to Theorem 5.2, such attacks will *never* affect consistency in OCERBERUS, however.

To further reduce the impact of targeted attacks, one can make primary election non-deterministic, e.g., by using shard-specific distributed coins to elect new primaries in individual shards [11, 13]. Finally, we remark that we have presented OCERBERUS with a per-round checkpoint and recovery method. In this simplified design, the recovery path only has to recover at-most a single round. Our approach can easily be generalized to a more typical multi-round checkpoint and recovery method, however. Furthermore, we believe that the way in which OCERBERUS extends PBFT can easily be generalized to other consensus protocols, e.g., POE [33] and HOTSTUFF [64].

# 6 Resilient-CERBERUS: Transaction Processing Under Attack

In the previous section, we introduced OCERBERUS, a general-purpose minimalistic and efficient multi-shard transaction processing protocol. OCERBERUS is designed with the assumption that malicious behavior is rare, due to which it can minimize coordination in the normal-case while requiring intricate coordination when recovering from attacks. As an alternative to the optimistic approach of OCERBERUS, we can apply a *pessimistic* approach to CCERBERUS to gracefully recover from concurrent transactions that is geared towards minimizing the influence of malicious behavior altogether (without requiring Assumption 2.6). Next, we explore such a pessimistic design via *resilient*-CERBERUS (PCERBERUS).

The design of PCERBERUS builds upon the design of CCERBERUS by adding additional coordination to the cross-shard exchange and decide outcome steps. As in CCERBERUS, the acceptance of $m(\mathcal{S},\tau)_{\rho}$ in round $\rho$ by all good replicas completes the *local inputs* step. Before cross-shard exchange, the replicas in $\mathcal{S}$ destruct the objects in $D(\mathcal{S},\tau)$, thereby fully pledging these objects to $\tau$ until the commit or abort decision. Then, $\mathcal{S}$ performs cross-shard exchange by broadcasting $m(\mathcal{S},\tau)_{\rho}$ to all other shards in shards($\tau$), while the replicas in $\mathcal{S}$ wait until they receive messages $m(\mathcal{S}',\tau)_{\rho'}$ from all other shards $\mathcal{S}' \in \text{shards}(\tau)$.

After cross-shard exchange comes the final *decide outcome* step. After $\mathcal{S}$ receives $m(\mathcal{S}',\tau)_{\rho'}$ from all shards $\mathcal{S}' \in$

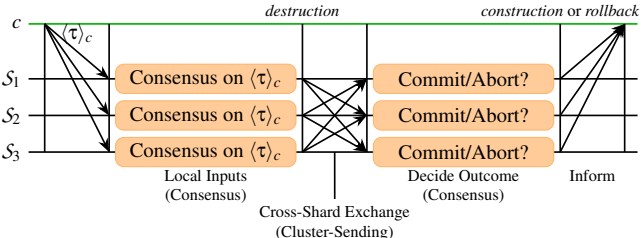

Figure 7: The message flow of PCERBERUS for a 3-shard client request $\langle\tau\rangle_c$ that is committed.

shards($\tau$), the replicas force a *second consensus step* that determines the round $\rho^*$ at which $S$ decides *commit* (whenever $I(S',\tau) = D(S',\tau)$ for all $S' \in$ shards($\tau$)) or *abort*. Note that the second consensus step in a shard $S$ starts after all good replicas in $S$ have received all required information to decide either commit or abort. Hence, the second consensus step is only used to *order* the commit or abort decision among all other decisions made in that shard. If $S$ uses a primary-backup consensus protocol, then this second consensus step can be initiated by the primary without further input. If the primary is malicious, the good replicas need to force the second consensus step, however (e.g., by replacing the primary).

If $S$ decides commit, then, in round $\rho^*$, all good replicas in $S$ construct all objects $o \in$ Outputs($\tau$) with $S =$ shard($o$). If $S$ decides abort, then, in round $\rho^*$, all good replicas in $S$ reconstruct all objects in $D(S,\tau)$ (rollback). Finally, each good replica informs $c$ of the outcome of execution. If $c$ receives, from every shard $S' \in$ shards($\tau$), identical outcomes from $\mathbf{g}_{S'} - \mathbf{f}_{S'}$ distinct replicas in $S'$, then it considers $\tau$ to be successfully executed. In Figure 7, we sketched the working of PCERBERUS.

We notice that processing a multi-shard transaction via PCERBERUS requires *two* consensus steps per shard. In some cases, we can eliminate the second step, however. First, if $\tau$ is a multi-shard transaction with $S \in$ shards($\tau$) and the replicas in $S$ accept $(\langle\tau\rangle_c, I(S,\tau), D(S,\tau))$ with $I(S,\tau) \neq D(S,\tau)$, then the replicas can immediately abort whenever they accept $(\langle\tau\rangle_c, I(S,\tau), D(S,\tau))$. Second, if $\tau$ is a single-shard transaction with shards($\tau$) = $\{S\}$, then the replicas in $S$ can immediately decide commit or abort whenever they accept $(\langle\tau\rangle_c, I(S,\tau), D(S,\tau))$. Hence, in both cases, processing of $\tau$ at $S$ only requires a single consensus step at $S$. Next, we prove the correctness of PCERBERUS:

**Theorem 6.1.** *If, for all shards $S^*$, $\mathbf{g}_{S^*} > 2\mathbf{f}_{S^*}$, and Assumptions 2.1, 2.5, and 2.6 hold, then Resilient-CERBERUS satisfies Requirements R1–R6.*

*Proof.* Let $\tau$ be a transaction. As good replicas in $S$ discard $\tau$ if it is invalid or if $S \notin$ shards($\tau$), PCERBERUS provides *validity* and *shard-involvement*. Next, *shard-applicability* follow directly from the decide outcome step.

If a shard $S$ commits or aborts transaction $\tau$, then it must have completed the decide outcome and cross-shard exchange steps. Hence, all shards $S' \in$ shards($\tau$) must have exchanged the necessary information to $S$. By relying on cluster-sending for cross-shard exchange, $S'$ requires cooperation of all good replicas in $S'$ to exchange the necessary information to $S$. Hence, we have the guarantee that these good replicas will also perform cross-shard exchange to any other shard $S'' \in$ shards($\tau$). Consequently, every shard $S'' \in$ shards($\tau$) will receive the same information as $S$, complete cross-shard exchange, and make the same decision during the decide outcome step, providing *cross-shard consistency*.

A client can force service on a transaction $\tau$ by choosing a shard $S \in$ shards($\tau$) and sending $\tau$ to all good replicas in $\mathcal{G}(S)$. By doing so, the normal mechanisms of consensus can be used by the good replicas in $\mathcal{G}(S)$ to force acceptance on $\tau$ in $S$ and, hence, bootstrapping acceptance on $\tau$ in all shards $S' \in$ shards($\tau$). Due to cross-shard consistency, every shard in shards($\tau$) will perform the necessary steps to eventually inform the client. As all good replicas $R \in S$, $S \in$ shards($\tau$), will inform the client of the outcome for $\tau$, the majority of these inform-messages come from good replicas, enabling the client to reliably derive the true outcome. Hence, PCERBERUS provides *service* and *confirmation*.  □

As with CCERBERUS, PCERBERUS depends on underlying consensus and cluster-sending protocols and the level to which PCERBERUS can deal with asynchronous behavior depends on the particular choices of these protocols.

The step from PCERBERUS to general-purpose workloads is small: one can replace the data model used by PCERBERUS by a general-purpose data model using two-phase locking instead of object construction and destruction to guarantee isolated execution. To guarantee the same level of isolation for transaction execution as PCERBERUS guarantees *without* using the UTXO-like transactions (see Section 7), such an approach will require a costly locking scheme for all pieces of data affected by a transaction (including any objects that would be considered outputs by PCERBERUS). Such a two-phase locking design will end up very similar to that of CHAINSPACE or BYSHARD.

## 7 The Ordering of Transactions in CERBERUS

Having introduced the three variants of CERBERUS in Sections 4, 5, and 6, we will now analyze the ordering guarantees provided by CERBERUS. We further refer to Section 8 for a detailed comparison of the three variants of CERBERUS. Here, we will show that CERBERUS provides serializable execution [6, 9].

The data model utilized by CCERBERUS, OCERBERUS, and PCERBERUS guarantees that any object $o$ can only be involved in at-most *two* committed transactions: one that

*constructs* $o$ and another one that *destructs* $o$. Assume the existence of such transactions $\tau_1$ and $\tau_2$ with $o \in \text{Outputs}(\tau_1)$ and $o \in \text{Inputs}(\tau_2)$. Due to *cross-shard-consistency* (Requirement R4), the shard $\text{shard}(o)$ will have to execute both $\tau_1$ and $\tau_2$. From these observations, we can derive a serializable order on all committed transactions:

**Theorem 7.1.** *A sharded fault-tolerant system that uses the object-dataset data model, processes UTXO-like transactions, and satisfies Requirements R1-R5 commits transactions in a serializable order.*

*Proof.* Assume the existence of transactions $\tau_1$ and $\tau_2$ with $o \in \text{Outputs}(\tau_1)$ and $o \in \text{Inputs}(\tau_2)$. Due to *shard-applicability* (Requirement R3), shard $\text{shard}(o)$ will execute $\tau_1$ strictly before $\tau_2$. Now consider the relation

$$\prec := \{(\tau, \tau') \mid (\text{the system committed to } \tau \text{ and } \tau') \wedge$$
$$(\text{Outputs}(\tau) \cap \text{Inputs}(\tau') \neq \emptyset)\}.$$

Obviously, we have $\prec(\tau_1, \tau_2)$. To prove that all committed transactions are executed in a *serializable* ordering, we first prove the following:

> If we interpret transactions as nodes and $\prec$ as an edge relation, then the resulting graph is *acyclic*.

The proof is by contradiction. Let $G$ be the graph-interpretation of $\prec$. We assume that graph $G$ is cyclic. Hence, there exists transactions $\tau_0, \ldots, \tau_{m-1}$ such that $\prec(\tau_i, \tau_{i+1})$, $0 \leq i < m-1$, and $\prec(\tau_{m-1}, \tau_0)$. By the definition of $\prec$, we can choose objects $o_i$, $0 \leq i < m$, with $o_i \in (\text{Outputs}(\tau_i) \cap \text{Inputs}(\tau_{(i+1) \bmod m}))$. Due to *cross-shard-consistency* (Requirement R4), the shard $\text{shard}(o_i)$, $0 \leq i < m$, executed transactions $\tau_i$ and $\tau_{(i+1) \bmod m}$. Consider $o_i$, $0 \leq i < m$, and let $t_i$ be the time at which shard $\text{shard}(o_i)$ executed $\tau_i$ and constructed $o_i$. Due to *shard-applicability* (Requirement R3), we know that shard $\text{shard}(o_i)$ executed $\tau_{(i+1) \bmod m}$ strictly after $t_i$. Moreover, also shard $\text{shard}(o_{(i+1) \bmod m})$ must have executed $\tau_{(i+1) \bmod m}$ strictly after $t_i$ and we derive $t_i < t_{(i+1) \bmod m}$. Hence, we must have $t_0 < t_1 < \cdots < t_{m-1} < t_0$, a contradiction. Consequently, $G$ must be acyclic.

To derive a serializable execution order for all committed transactions, we simply construct a directed acyclic graph in which transactions are nodes and $\prec$ is the edge relation. Next, we *topologically sort* the graph to derive the searched-for ordering. $\qquad \square$

We notice that CERBERUS only provides serializability for *committed* transactions: concurrent transactions that contend for the same objects will always be aborted and, hence, will not be executed and will not affect the serializable order of execution of transactions. This flexibility is crucial in the design of CCERBERUS: as a consequence of the minimal coordination in CCERBERUS, the order in which individual replicas make abort decisions is not coordinated. Consequently, replicas can end up processing abort decisions in distinct non-serializable orders, this even for replicas within a singular shard. It is this flexibility in dealing with aborted transactions that allows CCERBERUS to operate with minimal coordination while still providing strong isolation for all committed transactions. Both OCERBERUS and PCERBERUS incorporate stronger coordination, due to which all commit and abort decisions within a shard are made in a consistent order among all replicas in that shard. Still, both OCERBERUS and PCERBERUS do not enforce a particular order in which abort decisions are processed across distinct shards, as such enforcement would incur additional coordination costs.

## 8   Analysis of the Three CERBERUS Variants

In the previous sections, we proposed three variants of CER-BERUS and showed their correctness. Next, we analyze the benefits and costs of the three CERBERUS multi-shard transaction processing protocols, compare them with state-of-the-art multi-shard transaction processing protocols, and evaluate the impact of malicious behavior on CERBERUS. A summary of this analysis can be found in Figure 8.

We did not detail the exact message complexity of the three CERBERUS protocols. For CCERBERUS and PCERBERUS we measure the complexity in the number of consensus steps and cluster-sending steps they require. Implementation-wise, one can choose to either implement these steps with all-to-all communication (as in PBFT) or with all-to-one-to-all communication (as in HOTSTUFF) to optimize for either low latency or low bandwidth usage. Similarly, we can implement also OCERBERUS with all-to-one-to-all communication instead of all-to-all communication.

*Remark* 8.1. A common technique to improve the transaction throughput of consensus-based systems is by processing a batch holding many transactions per consensus decision. To simplify presentation, we have chosen to present the three CERBERUS protocols without such batching. Both CCER-BERUS and PCERBERUS can easily be generalized to process transactions in batches: at a per-shard level, they use standard consensus protocols that operate independently of other shards. Hence, instead of one transaction per consensus decision, both can include a batch of transactions in their consensus decisions (after which they perform the steps related to the transactions in that batch in order).

For OCERBERUS, blocks of transactions are more challenging as OCERBERUS uses a single multi-shard consensus step that includes all replicas of all shards affected by a transaction. Still, OCERBERUS can be generalized to process batches of transactions that affect the same set of shards. Such a generalization requires additional machinery, however: multi-shard batches can lead to several shards proposing distinct batches that include the same transaction, however. It is possible to

deal with such issues with existing techniques (e.g., by assigning each transaction to a single batch-proposal shard based on the digest of that transaction) [31].

## 8.1 A Comparison of CERBERUS Variants

First, Figure 8 provides a high-level comparison of the costs of each of the three CERBERUS protocols to process a single transaction $\tau$ that affects $s = |\mathsf{shards}(\tau)|$ distinct shards. For the normal-case behavior, we compare the complexity in the number of *sequential communication phases* (which, in the idle case, are the main determinant for client latencies), the number of *consensus steps* per shard and *cross-shard exchange* steps between shards (which together determine the bandwidth costs and put an upper bound on throughput). As one can see, all three protocols have a low number of *phases*, due to which all three can provide low latencies toward clients. On the one hand, we see that OCERBERUS only performs a PBFT-like multi-shard consensus, which takes *three* consecutive phases of communication. On the other hand, CCERBERUS performs both a local consensus step and a cross-shard exchange step. Assuming the local consensus step is performed via PBFT, this results in *four* consecutive phases of communication (three *local communication phases* for PBFT, one for the cross-shard exchange step). Hence, in environments in which cross-shard communication has low latency, OCERBERUS will be able to provide lower latencies than both CCERBERUS and PCERBERUS, as its optimistic design eliminates one phase of communication (at the cost of requiring cross-shard communication in every phase).

Next, we compare how the three protocols deal with malicious behavior by clients and by replicas. If no clients behave malicious, then all transactions will *commit*. In all three protocols, malicious behavior by clients can lead to the existence of concurrent transactions that affect the same object. Upon detection of such concurrent transactions, all three protocols will *abort*. The consequences of such an abort are different in the three protocols.

In CCERBERUS, objects affected by aborted transactions remain pledged and cannot be reused. In practice, this loss of objects can provide an incentive for clients to not behave malicious, but does limit the usability of CCERBERUS in non-incentivized environments. Both OCERBERUS and PCERBERUS deal with concurrent transactions by aborting them via the normal-case of the protocol. The three CERBERUS protocols are resilient against malicious replicas: only malicious primaries can affect the normal-case operations of these protocols. If the behavior of a primary is disrupting the normal-case operations, then in CCERBERUS and PCERBERUS such behavior is dealt with by the recovery mechanisms of the underlying consensus protocol (e.g., in PBFT such disruptions will eventually lead to a view-change whenever communication is reliable), whereas OCERBERUS will utilize the view-change recovery mechanisms outlined in Section 5. In both CCER-

BERUS and PCERBERUS, dealing with a malicious primary in a shard can be done completely in isolation of all other shards. In OCERBERUS, which is optimized with the assumption that failures are rare, the failure of a primary while processing a transaction $\tau$ can lead to view-changes in all shards affected by $\tau$.

In conclusion, we see that the three CERBERUS variants each make their own tradeoff between *normal-case costs* and ability to deal with faulty behavior (by both clients and other replicas), with PCERBERUS being robust against any attack at the cost of 2 consensus decisions per transaction per involved shard.

## 8.2 Comparison With the State-of-the-Art

Several recent papers have proposed specialized systems that combine sharding with consensus-based resilient systems. Examples include systems such as AHL [17], BYSHARD [36, 38], CAPER [3], CHAINSPACE [1], RINGBFT [58], and SHARPER [4], which all use sharding for data management and transaction processing. Next, we compare the design of CERBERUS in detail with AHL [17], CHAINSPACE [1], RINGBFT [58], and SHARPER [4], and briefly look at BYSHARD [36, 38] and CAPER [3].

**AHL [17].**   AHL uses a *centralized* commit protocol to order all multi-shard transactions. In specific, AHL [17] uses a reference committee that leads a *centralized two-phase commit protocol* (Centralized 2PC) [30, 56] that is implemented via consensus steps and cluster-sending. Typically, the reference committee is responsible for coordinating *all* multi-shard transactions. Furthermore, AHL uses non-blocking locks to provide transaction isolation due to which valid transactions can be aborted, whereas in CERBERUS only faulty transactions (e.g., by malicious clients) are aborted. By using Centralized 2PC, AHL eliminates any all-to-all communication between shards affected by a transaction in favor of one-to-all communication between the reference committee and the affected shards. Due to this, AHL takes five consecutive consensus rounds, more than twice the number of rounds required by the costliest CERBERUS variants. As reported in the original evaluation of AHL [17, Section 7.3], the central role of the reference committee in coordinating *all* multi-shard transactions will quickly become a bottleneck for performance when processing workloads heavy in multi-shard transactions (even if none of these transactions are concurrent), while AHL shows excellent performance when processing single-shard transactions [36, 38].

**CHAINSPACE [1].**   CHAINSPACE uses a *distributed two-phase commit protocol* (Distributed 2PC) [30, 56], that is implemented via consensus steps and cluster-sending, to order all multi-shard transactions. Furthermore, similar to AHL, CHAINSPACE uses non-blocking locks to provide transaction

| Protocol | Principle Technique | Phases[a] (Cross-Shard) | Consensus Steps. | | | Cross-Shard Communication[c] | Transaction Abort Causes | Transaction Concurrency and Ordering | Failure Recovery (method and when) |
| --- | --- | --- | --- | --- | --- | --- | --- | --- | --- |
| | | | Total | Sequential | Type[b] | | | | |
| CCERBERUS | Independent Consensus UTXO Data Model | 4 (1) | $s$ | 1 | LC | 1 (CS, A2A) | Faulty Only | Data Model Pledges (Incentive[d]) | Local Recovery Local Primary Failure |
| OCERBERUS | Multi-Shard Consensus UTXO Data Model | 3 (3) | $s$ | 1 | MS | 3 (MS, A2A) | Faulty Only | Data Model Aborts | Local and Global Recovery Any Primary Failure |
| PCERBERUS | Distributed Commit UTXO Data Model | 7 (1) | $2s$ | 2 | LC | 1 (CS, A2A) | Faulty Only | Data Model Aborts | Local Recovery Local Primary Failure |
| AHL [17] | Reference Committee Non-Blocking Locks | 19 (4) | $2s+2$ | 5 | LC | 4 (CS, O2A) | Failed Locks | Reference Committee Locks & Aborts | Local Recovery Local Primary Failure |
| CHAINSPACE [1] | Distributed Commit locking Locks | 11 (2) | $2s$ | 3 | LC | 2 (CS, A2A) | Failed Locks | Distributed Commit Locks & Aborts | Local Recovery Local Primary Failure |
| RINGBFT [58] | Linear Commit Blocking Locks | $8s-5$ $(2s-2)$ | $2s-1$ | $2s-1$ | LC | $2s-2$ (CS, O2O) | Invalid Only | Linear Commit Blocking Locks | Local Recovery Local Primary Failure |
| SHARPER [4] | Multi-Shard Consensus Shard-Wide Blocking Locks | 3 (3) | $s$ | 1 | MS | 3 (MS, A2A) | Failed Locks (Shard-Wide) | Multi-Shard Consensus Shard-Wide Locks & Aborts | Global Recovery Any Primary Failure, Concurrency |

[a]Total number of consecutive communication phases. For protocols that use a local consensus protocol, we count three consecutive phases per consensus step (e.g., using PBFT), and we count a single phase per cluster-sending step.

[b]We write *LC* to local per-shard consensus steps that do *not involve* cross-shard communication and we use *MS* to indicate multi-shard consensus steps that require cross-shard communication for each phase of communication performed by the multi-shard consensus protocol.

[c]We write *CS* to indicate *cluster-sending* and *MS* to indicate *multi-shard consensus*; and we write *A2A* to denote all-to-all communication, O2A to denote one-to-all or all-to-one communication, and O2O to denote one-to-one communication between involved shards.

[d]In CCERBERUS, objects affected by aborted transactions remain pledged and cannot be reused. In practice, this loss of objects can provide an incentive for clients to not behave malicious.

Figure 8: Comparison of the three CERBERUS protocols for processing a transaction that affects *s* shards. We compare the normal-case complexity. the mechanism used to deal with concurrent transactions (due to malicious clients), and the mechanisms used to provide failure recovery.

isolation due to which valid transactions can be aborted. The operations of this commit protocol are similar to the design of PCERBERUS, except that CHAINSPACE does not take advantage of any specific properties of the data model (e.g., to provide isolation). A further minor difference between CHAINSPACE and PCERBERUS is that CHAINSPACE distinguishes between shards that are used as inputs and shards that are used as outputs and only informs output shards after the input shards finish processing a transaction, due to which transaction processing in CHAINSPACE takes one more round as in PCERBERUS.

**RINGBFT [58].** RINGBFT uses a *linear two-phase commit protocol* (Linear 2PC) [30, 56], that is implemented via consensus steps and cluster-sending, to order all multi-shard transactions. Due to the usage of Linear 2PC, RINGBFT is able to utilize blocking locks in a deadlock-free manner to provide transaction isolation. Due to this usage of locks, RINGBFT is the only protocol besides CERBERUS that is able to always process valid transactions without spurious aborts. Furthermore, the usage of Linear 2PC minimizes cross-shard communication costs, as all communication is between pairs-of-affected-shards (no all-to-all, one-to-all, or all-to-one communication). The benefits of RINGBFT come at a cost, however, as the linear design imposes a *linear* amount of consecutive consensus and cross-shard communication steps in terms of the shards affected by the transaction, whereas all other proposals require a constant number of consecutive

steps.

**SHARPER [4].** SHARPER uses a *multi-shard consensus protocol* to order all multi-shard transactions. The operations of this multi-shard consensus protocol are conceptually similar to the design of OCERBERUS, except that SHARPER does not take advantage of any specific properties of the data model (e.g., to provide isolation or to simplify recovery). Furthermore, SHARPER requires that affected shards process their multi-shard transactions in a common processing order, In effect, this imposes a per-shard lock on multi-shard transaction processing, due to which each shard in SHARPER can only engage in at most a single multi-shard transaction at a time, limiting concurrent execution even in the absence of transactions that contend for the same data objects. This is in sharp contrast to the original design of PBFT on which both OCERBERUS and SHARPER are based: the design of PBFT enables implementations with an out-of-order design that allows for the processing of multiple consensus rounds at once, thereby enabling primaries to use all their available bandwidth to continuously propose transactions for future consensus rounds *without* waiting for previous consensus rounds to finish. This *out-of-order design* of PBFT enables it to process thousands of consensus rounds per second, even in real-world networks with high message delays [16, 31]. OCERBERUS *does* allow for out-of-order processing, however, as the correctness of OCERBERUS does not fundamentally depend on the order in which transactions that affect distinct objects are processed.

Finally, the philosophy of SHARPER is to serve as a single unified protocol that can support both PAXOS-style crash fault-tolerance and malicious behavior, and it remains an important research question as to whether SHARPER can be extended to the general-purpose unreliable communication and attack models supported by OCERBERUS. In specific, we believe OCERBERUS improves on the resilience of SHARPER by providing a *robust* local and global view-change mechanism that can deal with per-shard replica failures, per-shard primary failures, and coordinated attacks by replicas and clients to disrupt global consensus steps.

**BYSHARD [36, 38] and CAPER [3].** BYSHARD [36, 38] proposes a framework in which one can evaluate many distinct protocols based on the application of two-phase commit and two-phase locking in a consensus-based environment. Specific instances of BYSHARD correspond with the approaches taken by CHAINSPACE and RINGBFT, while AHL can be seen as a restricted case of the BYSHARD protocols that utilize distributed orchestration. The differences between, on the one hand, CERBERUS and, on the other hand, AHL, CHAINSPACE, and RINGBFT, extend to the BYSHARD framework. The design of CAPER [3] shares similarities with the design of SHARPER.

## 8.3 The Performance Potential of CERBERUS

Next, we modelled the performance benefits of CERBERUS. To do so, we have modeled the maximum throughput of each of these protocols in an environment where each replica has a bandwidth of 1 Gbit/s and the message delay is 15 ms.[4] Furthermore, we test with either seven (of which two can be faulty) or sixty-one replicas (of which twenty can be faulty). We have chosen to optimize CCERBERUS, OCERBERUS, and PCERBERUS to minimize *processing latencies* over minimizing bandwidth usage, as reducing processing latencies is the goal of the design of CERBERUS. In specific, we do *not* use request batching and we do not use *threshold signatures*. To further minimize latencies, we use a primary-initiated one-phase broadcast-based cross-shard exchange step that will succeed in a single communication phase whenever the sending primary is non-faulty and communication is reliable [32]. Although this cross-shard exchange step has low latency and can deal with failures, it does impose a high cost at the primary in the sending shard. In cases when one does not want to optimize for processing latencies and individual replicas have spare computational power, then one can utilize threshold signatures to further boost throughput by a constant factor (at the cost of the per-transaction processing latency).

In Figure 9, we have visualized the maximum attainable throughput (the number of transactions processed per second while processing a workload of 32M transactions) for each of the protocols as a function of the number of shards and as a function of the number of objects affected by each transaction when processing a workload in which 50% of the transactions are single-shard and the remaining transactions affect objects chosen fully-at-random. Hence, when multiple shards are present in the system, the remaining transactions are likely to be multi-shard transactions. As a baseline for comparison, we have also included CHAINSPACE [1] and AHL [17]. For AHL, we used an additional shard as a reference committee (hence, if we use $n$ shards in the experiment, then AHL can use $n + 1$).

In general, we see that CCERBERUS has higher throughput than OCERBERUS and OCERBERUS has higher throughput than PCERBERUS. These results are easily explained: all protocols are bottlenecked by the communication at the primaries of shards (as these primaries broadcast proposal messages, which are much larger than any other messages), and in general the shards perform the least amount of work per transaction in CCERBERUS and the most amount of work per transaction under PCERBERUS.

For example, when comparing CCERBERUS with OCERBERUS, we see that the multi-shard consensus of OCERBERUS makes a trade-off in favor of latency at the expense of *higher communication costs* at the primaries of shards than the higher-latency combination of local consensus and cross-shard exchange used by CCERBERUS. This is not always true, however. To minimize latencies for CCERBERUS and PCERBERUS, we used a primary-initiated one-phase broadcast-based cross-shard exchange step that imposes a high cost at the primary in the sending shard. As a consequence, we see that multi-shard transactions that only affect a few shards (e.g., very few shards or very small transactions), result in less communication at the primaries under the multi-shard consensus performed by OCERBERUS than the combined cost of a local consensus and cross-shard exchange step performed by PCERBERUS. Consequently, we see slightly higher throughput for OCERBERUS than for CCERBERUS in these settings.

For RINGBFT, we notice that it is optimized for a completely different design goal: RINGBFT is optimized to minimize communication costs and maximize throughput (by inducing very high latencies for multi-shard transactions), whereas the three CERBERUS protocols aim to maximize throughput while also minimizing latency. To further underline this, we measured the *minimal latency* of processing a multi-shard transaction as a function of the number of affected shards for the three CERBERUS protocols and for RINGBFT, CHAINSPACE, and AHL. We refer to Figure 10 for the results. As is clear from the results, only the latency of RINGBFT is strongly determined by the number of shards. This is not a surprise, as the three CERBERUS protocols, CHAINSPACE,

---

[4]We have chosen for a bandwidth of 1 Gbit/s and a message delay of 15 ms as this message delay is multiple orders of magnitudes larger than the time it takes to send all bytes of any individual message. Hence, if the protocol has bottlenecks based on message delay, then these bottlenecks will clearly show with the parameters chosen.

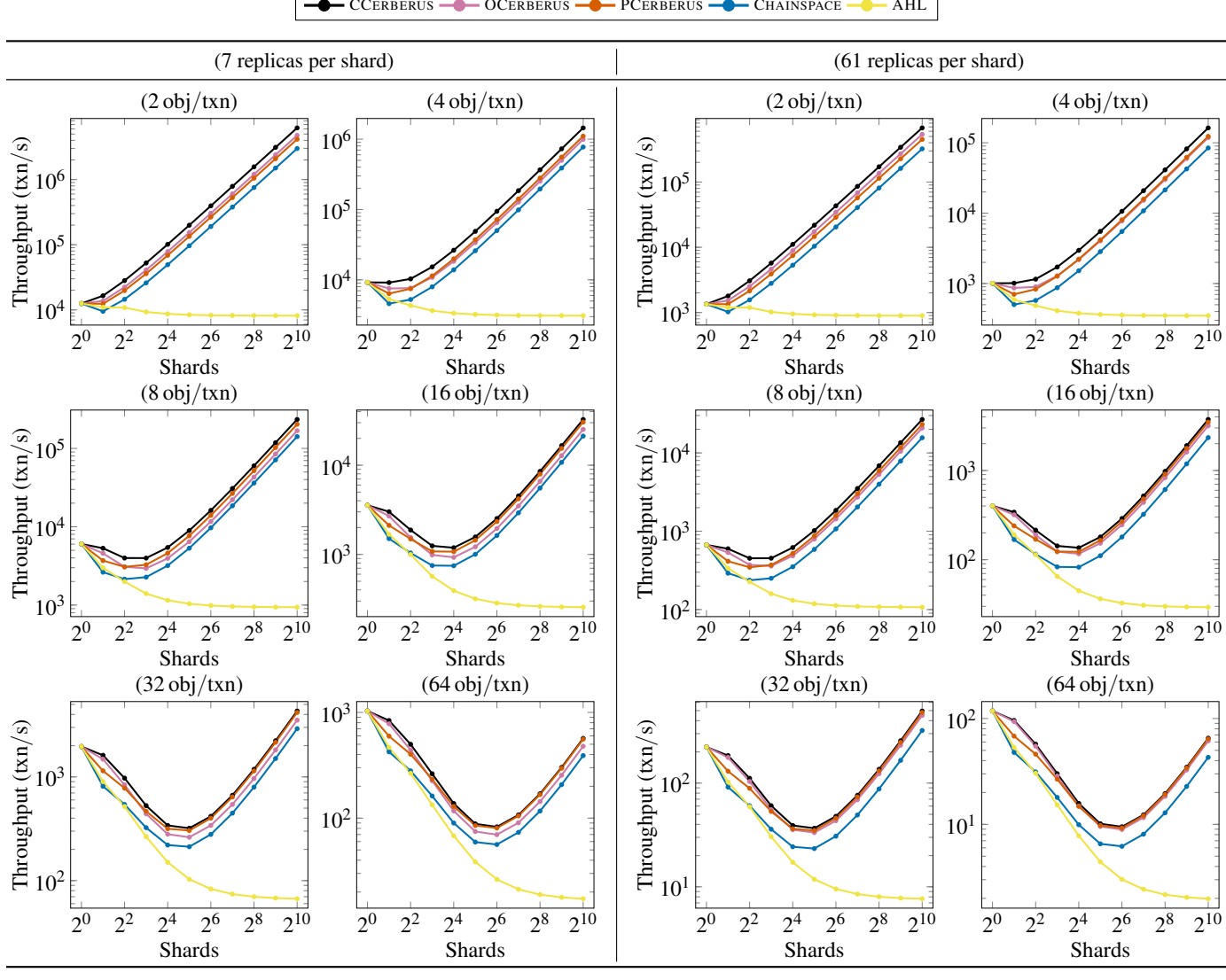

Figure 9: Throughput of the three CERBERUS protocols as a function of the number of shards when processing a workload in which 50% of the transactions are single-shard transactions and all remaining transactions affect objects chosen fully-at-random and, hence, are likely to be multi-shard transactions.

and AHL all operate in a fixed number of consecutive steps, whereas in RINGBFT the number of consecutive steps is a function of the number of affected shards. In specific, CCER-BERUS has two consecutive steps (a consensus and a cross-shard exchange step); OCERBERUS has a single step (a multi-shard consensus step); PCERBERUS has three consecutive steps (two consensus steps and a cross-shard exchange step); CHAINSPACE has five consecutive steps (three consensus steps and two cross-shard exchange steps); and AHL has nine consecutive steps (five consensus steps and four cross-shard exchange steps). With regards to behavior, CHAINSPACE is similar to PCERBERUS, except that the construction of outputs is a separate step that happens *after* inputs are destructed. The high number of consecutive steps in AHL is a design

choice aimed at reducing *overall* inter-shard communication at the cost of latency by *not* using all-to-all broadcasts between affected shards, but instead steering all communication between shards via an all-to-*reference committee*-to-all communication pattern, which introduces additional consensus steps.

In our workloads, the *ratio of multi-shard transactions* is high: we want to study how the multi-shard transaction processing protocols we compare differ in their operations and we are especially interested in the performance of the system when dealing with *multi-shard transactions* that require substantial coordination to deal with contention. Indeed, in workloads that mainly consist of *single-shard transactions*, each of the multi-shard transaction protocols we look at will

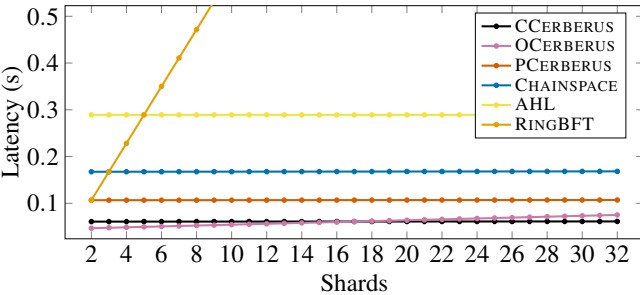

Figure 10: Latency of processing a transaction during multi-shard transaction processing as a function of the number of affected shards, assuming that the transaction affects 64 objects, the network has a bandwidth of $1\,\text{Gbit/s}$, the message delay is $15\,\text{ms}$, and each shard has 7 replicas.

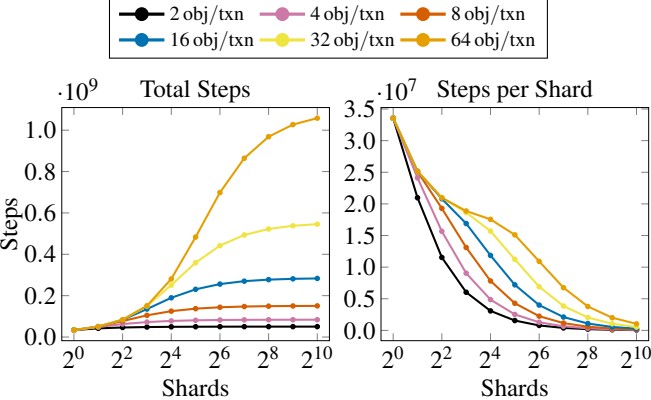

Figure 11: Cumulative number of shards affected by a workload of 32M transactions as a function of the size of transactions and the number of shards. On the *left*, the *total* number of steps. On the *right*, the *average* number of steps per shard. For CCERBERUS and OCERBERUS, the number of steps is equivalent to the number of per-shard consensus steps, for PCERBERUS the number of steps is *half* the number of per-shard consensus steps.

fall back to the same underlying single-shard consensus protocol to efficiently process such single-shard transactions.

Finally, we note that the results in Figure 9 show a V-shape: when moving from an unsharded system to a sharded system with *few* shards, the throughput decreases, whereas moving to *many* shards will yield a sharp increase in throughput. This shape is a consequence of the very high ratio of multi-shard transactions in our workload: when going from an unsharded system to a sharded system with a *few* shards, a huge fraction of transactions affects most (or even all) shards: all these transactions enforce extra coordination costs, with no scaling benefit. As each transaction only affects $i$ objects, $i \in \{2,...,64\}$, each transaction can affect at-most $i$ shards. As such, scaling beyond $i$ shards will always show scaling benefits.

In Figure 11, we have visualized the number of per-shard steps performed by the system (for CCERBERUS and OCERBERUS, this is equivalent to the number of per-shard consensus steps, for PCERBERUS this is half the number of per-shard consensus steps). In general, we see that an increase in shards has two effects:

1. Simple single-shard transactions can be dispersed over more shards. Hence, increasing the number of shards will reduce the average number of shard steps each shard has to process with respect to *single-shard transactions*. Furthermore, significantly increasing the number of shards will distribute the *multi-shard transactions* over these shards, reducing the cost of these transactions per shard. Both effects will result in drastically improved performance when scaling to many shards.

2. Large transactions can become more complex when increasing the number of shards, as more shards can hold objects relevant of large transactions. For example, a transaction that affects 16 objects in an environment with four shards can affect at-most four shards, while in an environment with 16 shards it can affect at-most 16 shards). Hence, for large transactions, we only see reductions in the per-shard cost to process these transactions when scaling beyond the number of shards large transactions can affect.

In a general-purpose sharded system without any specific bottlenecks, the above will result in great scalability as soon as the number of shards far outgrows the size of transactions. This behavior is clearly observable for all three CERBERUS protocols. Indeed, all three CERBERUS protocols have excellent scalability: increasing the number of shards will increase the overall throughput of the system. Sharding does come with clear overheads, however, increasing the number of shards also increases the number of shards affected by each transaction, thereby increasing the overall number of consensus steps. This is especially true for very large transactions that affect many objects (that can affect many distinct shards). Hence, as one can see from the results, the benefits of sharding are the strongest when processing mainly single-shard transactions or when scaling beyond the size of individual transactions.

In the comparison between OCERBERUS and PCERBERUS, we see that OCERBERUS (implemented with all-to-all communication) will outperform PCERBERUS whenever transactions involve few shards (due to them involving few objects). In this case, the communication cost of the three cross-shard communication steps that are part of the multi-shard consensus of OCERBERUS is lower than the cost of the second local consensus round in PCERBERUS. When transactions affect many shards, PCERBERUS outperforms OCERBERUS, as PCERBERUS only has a single cross-shard communication step per transaction (and all other communication is local

within a shard). In all cases, CCERBERUS will outperform the other protocols with respect to transaction throughput. Furthermore, by design CCERBERUS will always have lower latencies than PCERBERUS (due to the fewer consensus and cluster-sending steps CCERBERUS performs). In environments in which inter-shard and intra-shard communication have similar (high) message delays, OCERBERUS will typically have lower latencies than CCERBERUS due to the large impact message delays have on the latency of consensus and cluster-sending steps, this even when CCERBERUS has higher throughput than OCERBERUS.

In comparison with CHAINSPACE, we see that this protocol behaves both in theory and in practice similar to PCERBERUS: the main difference being that CHAINSPACE commits in two steps (first, changes to the inputs are committed, then, in a consecutive consensus step, the changes to the outputs are committed). Due to this, CHAINSPACE has slightly higher costs associated with processing multi-shard transactions than PCERBERUS.

In comparison with AHL, we see a large improvement in performance. Due to the high ratio of multi-shard transactions, the performance of AHL for processing multi-shard transactions is bottlenecked by the throughput of the reference committee used by AHL to coordinate processing of *all* multi-shard transactions. These findings are in line with the original evaluation of AHL [17, Section 7.3]. The usage of a reference committee is contrasted by the design of the CERBERUS protocols: the CERBERUS protocols do *not* depend on an external coordinator to process multi-shard transactions. Instead, in the CERBERUS protocols, processing a multi-shard transaction $\tau$ is coordinated *only* by the shards affected by $\tau$, due to which the CERBERUS protocol can concurrently process multi-shard transactions that affect distinct sets of shards without introducing bottlenecks. A closer look at the data does reveal *excellent* scalability of AHL with regards to single-shard transactions: although the reference committee has a full load while processing all multi-shard transactions, all shards *except* the reference committee show a very low load (that can be used to process many more single-shard transactions during the experiment).

## 8.4 CERBERUS and Malicious Behavior

Finally, we have modeled the maximum throughput of each of the three CERBERUS protocols in an environment in which some shards are impacted by malicious replicas. Unless stated otherwise, we use the same setting as in Section 8.3. We have used 64 shards and four objects per shard and we measure the performance of the three CERBERUS protocols as a function of the number of shards that are affected by malicious behavior.

Within a shard, only malicious primaries have a strong impact on the performance of that shard. Furthermore, malicious primaries that completely disrupt normal-case operations will

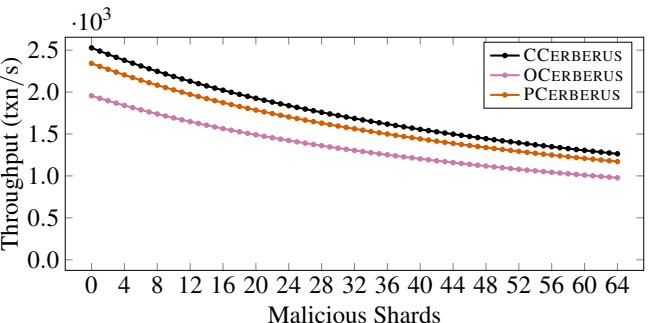

Figure 12: Throughput of the three CERBERUS protocols as a function of the number of shards that are affected by malicious behavior, as measured by the transaction cost at each shard. We have 16 obj/txn for all transactions. In this case, the shards affected by malicious behavior are controlled by malicious primaries that throttle the throughput of that shard to *half* the maximum attainable throughput.

be replaced. Hence, to maximize the malicious impact, we have chosen for malicious primaries that try to *throttle* the performance of the system by slowing down their own operations. In the experiment, we have chosen that these primaries do so by halving the speed by which their shards operate.

In Figure 12, we have visualized the average attainable throughput for shards processing workloads in which 50% of the transactions are multi-shard as a function of the number of shards that are affected by malicious behavior (and can affect the speed by which some of the multi-shard transactions processed are processed) for each of the CERBERUS protocols. The shards affected by malicious behavior are set up with malicious primaries that purposely throttle the throughput of the system by slowing down their operations to *half* the speed they could attain. We see that each shard affected by malicious behavior will slow down each CERBERUS protocol with respect to those transactions that are affected by these shards. At the same time, transactions not handled by shards affected by malicious behavior shards are unaffected and will be processed at normal speed due to which the impact of a few affected shards is minimal.

## 9 Related Work

Distributed systems are typically employed to either increase reliability (e.g., via consensus-based fault-tolerance) or to increase performance (e.g., via sharding). Consequently, there is abundant literature on such distributed systems, distributed databases, and sharding (e.g., [56, 60, 61]) and on consensus-based fault-tolerant systems (e.g., [10, 14, 19, 31, 60]). Furthermore, in Section 8.2, we reviewed related work on multi-shard permissioned consensus-based systems. Next, we focus on other works that deal with sharding in fault-tolerant systems.

In Section 8.2, we have only compared with other sharded

resilient systems with similar environmental assumptions. Besides these sharded systems, several other resilient systems such as OMNILEDGER [45] and RAPIDCHAIN [66] have been proposed. These systems make very different environmental assumptions (e.g., different threat and communication models) due to which these systems are incomparable to the CERBERUS protocols and the systems considered in Section 8.2.

A few fully-replicated consensus-based systems utilize sharding at the level of consensus decision making, this to improve consensus throughput *without* adopting a multi-shard design [2, 22, 26, 32]. In these systems, only a small subset of all replicas, those in a single shard, participate in the consensus on any given transaction, thereby reducing the costs to replicate this transaction without improving storage and processing scalability.

Recently, there has also been promising work on sharding and techniques supporting sharding for permissionless blockchains. Examples include techniques to enable sidechains, blockchain relays, and atomic swaps [23, 24, 39, 41, 46, 63, 65], which each enable various forms of cooperation between blockchains (including simple cross-chain communication and cross-chain transaction coordination). Unfortunately, these permissionless techniques are several orders of magnitudes slower than comparable techniques for traditional fault-tolerant systems, making them incomparable with the design of CERBERUS discussed in this work.

## 10 Conclusion

In this paper, we took a new look at the problem of multi-shard transaction processing in consensus-based systems. In specific, we proposed the study of *sharded consensus-based systems* that use restrictions on the workloads supported to improve performance over general-purpose methods.

To initiate this study, we introduced Core-CERBERUS, Optimistic-CERBERUS, and Resilient-CERBERUS, three fully distributed approaches towards multi-shard fault-tolerant transaction processing. The design of these approaches is geared towards processing UTXO-like transactions in sharded distributed ledger networks. Due to the usage of UTXO-like transactions, the three CERBERUS variants can minimize cost to an absolute minimum, while maximizing performance, thereby showing the potential of *restricting the types of supported workloads*. This potential is further underlined by our comparison with the state-of-the-art protocols, in which we see that the three CERBERUS variants both have lower costs and complexity.

Although the workloads supported by CERBERUS are minimalistic, we believe that our results can be generalized to more-general settings. In specific, we believe that the combination of sharding and *Conflict-free Replicated Data Types* (CRDTs) [49] has great potential to provide high performance in a consensus-based environment. Another intriguing direction is the investigation of resilient systems that *not only* limit

the types of workloads supported, *but also* reduce the resilient guarantees provided by the system, this to further maximize throughput and minimize latency. For example, by no longer using a consensus-based design but instead using small quorum systems [50].

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
