# OpenReview forum: "Problem: Cerberus: Minimalistic Multi-shard Byzantine-resilient Transaction Processing"
_JSYS/2023/March_Papers — Accept (with shepherding)_

### Official Review · Reviewer_n4H1 · 2023-04-07

**Decision:**

Strong accept: excellent paper that will help the community

**Strengths:**

* The problem now is well-defined. Assumptions and Definitions of the workload and the underlying consensus protocol were refined. Also, terminology is now consistent and accurate across the paper.
* The Evaluation section improved significantly. Results for CHAINSPACE and AHL are now included in all figures. Also, the current Evaluation section has better explanation and analysis of the results.


**Weaknesses:**

* The use of UTXO transactions is not well motivated. I suggest defining UTXO transactions and highlighting the difference between them and general transactions in the Introduction. Maybe moving footnote 2 (page 3) to the Introduction.

* Nits
	* into a object --> intro an object (page 3)
	* we say that a transaction received by the system is discarded if it will never be processed --> I guess “executed” instead of “processed” (page 4)
	* consensus decision --> consensus decisions (page 7)


**Expertise:**

Follow the literature closely, last published 5+ years ago

**Summary Of Review:**

The revision addresses most of the comments in the reviews from the previous round. The problem now is well-defined and the used terminologies across the paper are consistent. Also, the Evaluation section improved significantly; it includes results of other alternatives and has a better explanation of the results.

**Useful:**

yes

---

### Official Review · Reviewer_atuE · 2023-04-10

**Decision:**

Weak accept: good paper with flaws that can be fixed in three months

**Strengths:**

* Improved problem definition so that it’s easier to understand

* More comparison with related systems so that it’s more clear on the benefits of the proposed frameworks

* Narrowing the scope of the targeting systems which helps readers build a more clear mental model


**Weaknesses:**

* The presentations can be improved at several places (see below)

* The evaluation is purely analytical, and is only on a single environment (more below)



**Detailed Comments:**

Thank you for addressing prior comments with great details and effort. I can now understand and appreciate the problem a bit more.

Major points:

The authors assume that the readers are very familiar with the literature. Here are a few example in Section 1:
1. “Can one reduce the cost of coordination in the design of sharded consensus-based systems by limiting the types of workloads supported?” →  Would it be more clear to say that by considering only UTXO-like transactions and object models, instead of limiting the types of workloads?
2. At P2, it’s not clear what “local” means before presenting the key contributions, e.g., “​​UTXO-transactions to its advantage to yield a minimalistic design that only requires a single local consensus step…”
3. At P2, it’s not clear what recovery path and localized mean in “CCHIMERA to enable a simpler and localized recovery path,”
4. What exactly is UTXO-like transactions? How is it different from UTXO used in Bitcoin? More clarification will be needed.

The contribution of Optimistic-CHIMERA should be made more clear by mentioning the assumptions of optimistic case (i.e., Proposition 5.3).

I’m not a big fan of excessive use of many long footnotes. I feel they disrupt the reading quite a bit.

The description of the object-dataset model is still not clear. I had to read the responses to my previous comments to understand what the model is intended to do. Consider addressing the following points:
1. Give an example of its usage
2. Clearly identify the difference between object-dataset and object storages (e.g., key-value storage). I believe Footnote 2 is aiming to do so. However, Footnote 2 is not quite precise in the sense that in a versioned file system, an old file doesn’t need to be destroyed. And multiple operations can be operated on a single version of the file simultaneously, which is vastly different from the proposed model.
3. I assume there is no reading of the object? Since the only supported operations are create and remove?
4. Clearly mention that the approve of the owner is already embedded in the transaction
5. Clearly define what “consumes” mean
6. While it’s not typical, I think it makes sense for the owner of the output object to “approve” a transaction. For example, I need to approve a transaction for another to send me a token or NFT. Without such approval, it seems to me that anyone can send any NFT to my wallet without my consent?

Personally, I’m not sure if it’s the best idea to discuss PBFT and view-change right after Def. 3.1. This seems to disrupt the flow. Is there a better way to explain property 3 and its implication?

A better explanation on how the figures are generated is necessary. For example, I still don’t quite understand the v-shape of Figure 9. Is it because of the ratio between single-shard and multi-shard operations?

It will greatly improve the paper if more setups are considered in evaluation. For example, varying message latency (at least some setup that shows the effect of local consensus vs. cross-shard consensus). If possible, a different way of choosing shards (instead of uniformly random) will be useful.





Minor points:
P1: “ an absolute minimum amount of coordination between shard” → Maybe just write it out what exactly is needed?

P1: “CCHIMERA will operate perfectly with respect to all trans- actions” → not clear what perfectly means here

P1: “but does not provide any other guarantees” → Not clear

P1: “E.g., in Ethereum, a popu- lar public permissionless DLT …” → Consider replacing E.g., with For example

P1: the third paragraph is really long. Consider shortening it.

P1: “As such, these solutions lack the scalability required by many modern data-based applications.” → Consider using data-intensive applications. Otherwise, data-based applications is not clear. As one can argue, any application is based on some sort of data.

P3: It’s not clear what  “coordinating adversaries” is. What’s wrong with just using adversaries?

P4: “that a shard S performs an action if every good replica in G (S ) performs that action.” → “eventually” performs that action?

P4: “ reliable ordered replication of decisions (e.g., the decision to execute a given transaction next).” → What is a reliable ordered application? What is transaction next?

P4: “ it can force consensus on D” → The notion of force is not clear to me.

P4: “during which all preceding consensus decisions are secured” → secured is not clear here

P4: The notion of dataset is not clear in R3

P4: What does it mean for a transaction to be shard-applicable? R3 is a property for the system, not transaction itself

P5: “the shard S to propose a consensus decision in some consensus round ρ” → the notion of consensus round is not clear here

P13: consider putting the extension of RCHIMERA into a standalone subsection, as it’s an important point.

P17: What CAPER is not discussed along with SHARPER if they’re similar?


**Expertise:**

Actively publishing in this area

**Summary Of Review:**

This version of the paper has improved after addressing many comments from the previous round. In particular, the problem definition and the evaluation parts are more clear; hence, I believe that the proposed problem could be beneficial for the community. While as noted by other reviewers, the proposed solutions are not technically novel, I do see the values in formulating the problems and the insights from the preliminary analysis. That being said, there are still many places to be fixed so that the more general system audience could appreciate the work.

**Useful:**

yes

---

### Official Review · Reviewer_Xoa9 · 2023-04-12

**Decision:**

Strong accept: excellent paper that will help the community

**Strengths:**

- problem is relevant to the community
- problem seems relatively important
- formalization of "sharded consensus-based system" seems useful, at least in the "UTXO-like" restriction.
- CChimera is indeed "optimal" under their definitions

**Weaknesses:**

- I am still unclear on the purpose of the AHL comparison. If your goal is to prove that limiting workload type can improve performance, but AHL's performance is "overshadowed by the peculiarities of the design of AHL," why bother talking about AHL at all? What does it prove?
- It is not clear to me that the "sharded consensus-based system" formalization accurately reflects the designs used in sharded blockchains. For example, Polkadot and Eth2 both use a central "beacon" chain, which actually executes consensus, to checkpoint all the shards (which need no consensus of their own). That said, the authors provide examples of systems that satisfy this formalization in literature, so they're definitely worth talking about.
- I'm not clear on the claim "our protocols provide stronger guarantees on the status of individual good replicas in all shards (due to the usage of consensus)." What guarantees would CChimera be able to make on individual good replicas that, say, Small Byzantine Quorums (now mentioned in the end of the conclusion) could not make? It seems to me that, in either case, the only inconsistencies individual replicas can have are when "bad" resource owners submit conflicting transactions: CChimera can end up with some but not all of the transaction's inputs consumed, and SBQ can end up with inconsistent timestamps written to some registers, effectively making them unreadable. These seem like fairly similar failures. Am I missing something? If so, is this something other readers might also miss, which might be worth clarifying?

**Detailed Comments:**

The last paragraph in section 6 now clarifies that the relative advantage of RChimera over Chainspace or ByShard may stem from the workload restriction: UTXO-like transactions. It would be interesting to see a more direct demonstration (e.g. an actual variation of RChimera without the workload restriction for performance comparison), but I do not think that is reasonable to ask for publication.

The same paragraph would seem to apply (partially) to OChimera as well: OChimera is also restricted to UTXO-like transactions. Perhaps the wording in the introduction can be tweaked to make it clear that "UTXO-like" is one of the performance-enhancing restrictions, and not simply a way of formatting transactions. For example, the phrase "we show that all three variants of Chimera provide strong ordering guarantees based on their usage of UTXO-transactions" could be amended to something like "we show that restricting our transactions to the UTXO framework grants performance benefits to all three Chimera variants, even with strong ordering guarantees."


Nits:

Should the phrase "even replicas within a singular shard" be "includes replicas in the same shard" ?

I'm not sure the phrase "both OChimera and RChimera do not enforce a distinct non-serializable order in which abort decisions are processed across shards" is correct. Did you mean "distinct serializable order" ?



**Expertise:**

Actively publishing in this area

**Summary Of Review:**

This paper presents a problem new to the JSys community, specifically: "Can one reduce the cost of coordination in the design of sharded consensus-based systems by limiting the types of workloads supported?" The authors define a "sharded consensus-based system," and demonstrate that coordination costs can be reduced in practice using their own "Chimera" family of protocols, with comparison to previous protocols for unlimited workloads.

The paper argues that this problem is important, as it impacts the design of performance-sensitive sharded blockchains, which may implement sharded consensus-based systems. Such blockchains may want to limit their workloads in order to improve performance using protocols like Chimera.

This revision has addressed almost all of the comments from my previous review.

**Useful:**

yes

---

### Meta-Review · Area_Chair_kyew · 2023-04-19

**Recommendation:** Accept
**Confidence:** 5

**Metareview:**

Dear Authors,

Thank you again for submitting the revised version of your manuscript titled "Sharded Resilient Transaction Processing with Minimal Costs". Reviewers have agreed to accept your paper. Congratulations! Thank you for the hard work you have put on producing this version.

As per JSys rules, your paper will undergo shepherding to produce the final camera ready version. Although all reviewers showed excitement in accepting the paper, there are still many comments we would like to be addressed in the final version of the paper. Please follow the deadlines below to ensure timely shepherding and preparation of the camera ready version:

1) We kindly ask you to look at the reviews and prepare a list of changes that you plan to make in the final version of the paper. Please share that list with us in a week or so (around Tuesday, April 25th). We will approve the list asap.
2) Please submit the revised version, with annotated changes, by May 10, 2023, or earlier of course. The revision needs to follow the approved list.
3) The camera ready version is due in one month, May 18, 2023 (hard deadline).

If you have any questions or comments, please do not hesitate to reach out to us. Thank you again for submitting to JSys, and congratulations!

Lewis Tseng and Roberto Palmieri
Area chairs of JSys

---

### Decision · Program_Chairs · 2023-04-27

**Decision:**

Accept (with shepherding)

**Comment:**

Congratulations on getting your manuscript accepted!

The meta-review contains details about the reviewers' expectations for the final version. Please reach out to the Area Chairs if anything is unclear.

Romain Jacob, JSys Editor-in-Chief